# IL-9 aggravates SARS-CoV-2 infection and exacerbates associated airway inflammation

Srikanth Sadhu[1,2], Rajdeep Dalal[1], Jyotsna Dandotiya[1], Akshay Binayke[1], Virendra Singh[1], Manas Ranjan Tripathy[1,2], Vinayaka Das[1], Sandeep Goswami[2], Shakti Kumar[3], Zaigham Abbas Rizvi[1,2] & Amit Awasthi [1,2] ✉

SARS-CoV-2 infection is known for causing broncho-alveolar inflammation. Interleukin 9 (IL-9) induces airway inflammation and bronchial hyper responsiveness in respiratory viral illnesses and allergic inflammation, however, IL-9 has not been assigned a pathologic role in COVID-19. Here we show, in a K18-hACE2 transgenic (ACE2.Tg) mouse model, that IL-9 contributes to and exacerbates viral spread and airway inflammation caused by SARS-CoV-2 infection. *ACE2.Tg* mice with CD4[+] T cell-specific deficiency of the transcription factor Forkhead Box Protein O1 (Foxo1) produce significantly less IL-9 upon SARS-CoV-2 infection than the wild type controls and they are resistant to the severe inflammatory disease that characterises the control mice. Exogenous IL-9 increases airway inflammation in Foxo1-deficient mice, while IL-9 blockade reduces and suppresses airway inflammation in SARS-CoV-2 infection, providing further evidence for a *Foxo1*-Il-9 mediated Th cell-specific pathway playing a role in COVID-19. Collectively, our study provides mechanistic insight into an important inflammatory pathway in SARS-CoV-2 infection, and thus represents proof of principle for the development of host-directed therapeutics to mitigate disease severity.

Severe acute respiratory syndrome coronavirus 2 (SARS-CoV-2) causes coronavirus disease 2019 (COVID-19). COVID-19 symptoms range from mild to severe pneumonia and acute respiratory distress syndrome[1]. SARS-CoV-2 infection leads to hyperactivation of immune cells, which further induces inflammatory cascade, broncho-alveolar inflammation and immunopathology[2]. Use of dexamethasone, an anti-inflammatory drug, resulted in lower mortality and severity in patients hospitalized with COVID-19[3], indicating that immune suppression is effective in controlling the severity and mortality. In addition, we identified that SARS-CoV-2 infection, in animal model, contributes to the extra-pulmonary pathologies which include cardiovascular complications and thymic atrophy[4].

Although the precise mechanism of disease pathogenesis and lung pathology that lead to hyper-inflammatory response is not fully understood, autopsy histopathology of pulmonary samples revealed increased accumulation of eosinophils, basophils, neutrophils and perivascular and septal mast cells in COVID-19[5–7]. Single-cell landscape of bronchoalveolar immune cells in patients with COVID-19 shows that one of the prominent cell types is Mast cells[7,8]. In line with this, mast cells-derived proteases, chymase and eosinophil-associated mediators are found to be elevated in sera of COVID-19 patient and lung autopsies[5,8]. Interleukin 9 (IL-9), a common γ chain family cytokine primarily produced by Th9 cells[9], promotes mast cell growth and function in allergic inflammation[10]. Although IL-9 plays an essential role in severe airway inflammation and bronchial hyper responsiveness in asthma and Respiratory Syncytial Virus (RSV) infection[11,12], the role of IL-9 is not yet identified in SARS-CoV-2 infection, and its associated immunopathology.

[1]Centre for Immunobiology and Immunotherapy, Translational Health Science and Technology Institute, NCR-Biotech Science Cluster, 3rd Milestone, Faridabad 121 001 Haryana, India. [2]Immunology-Core Laboratory, Translational Health Science and Technology Institute, NCR-Biotech Science Cluster, 3rd Milestone, Faridabad 121 001 Haryana, India. [3]Centre for Human Microbiome and Anti-Microbial Resistance, Translational Health Science and Technology Institute, NCR-Biotech Science Cluster, 3rd Milestone, Faridabad-Gurgaon Expressway, Faridabad 121001 Haryana, India. ✉e-mail: aawasthi@thsti.res.in

IL-9 induction in Th9 cells requires a distinct set of transcription factors. *Pu.1* is one of the key transcription factors essential for IL-9 induction and Th9 cell differentiation[9]. Other transcription factors, Interferon (IFN) regulatory factor 1 (*Irf-1*)[13], *Irf-4*[14], Basic leucine zipper transcription factor, ATF-like (*Batf*)[15] and Hypoxia-inducible factor 1-alpha(*Hif-1α*)[16,17], were found to be associated with the Th9 cells differentiation and functions. We have identified that *Foxo1*, a forkhead family transcription factor, is essential for IL-9 induction not only in Th9 cells but other Th subsets[18]. In addition, the role of *Foxo1* is well documented in T cell homoeostasis and tolerance through interplay between effector and regulatory T cell[19]. While *Foxo1* regulates Th17[20–22] cell functions, it is essential for the generation and functions of Foxp3+ Tregs and Th9 cells[23].

This study aims to investigate the role of interleukin-9 (IL-9) in COVID-19 pathogenesis. We show that hACE2.Tg mice with CD4+ T cell-specific deficiency of the transcription factor *Foxo1* are resistant to severe inflammation. Further, we also observe that exogenous IL-9 increases airway inflammation in Foxo1-deficient mice, while IL-9 blockade reduces and suppresses airway inflammation in SARS-CoV-2 infection. Here we unravel the role of Foxo1-IL-9 axis in SARS-CoV-2 infection, and its associated airway inflammation and immunopathology. These findings provide important mechanistic insights in understanding role of Foxo1-IL-9 axis in COVID-19 and could pave the way for the development of host-directed therapeutics to mitigate respiratory viral illness and disease severity.

## Results

### IL-9 promotes the severity of SARS-CoV-2 infection and associated lung inflammation

We show that IL-9 levels were elevated in active COVID-19 patients infected with ancestral strain during the first wave of pandemic in mid-2020 (Fig. 1a; Supplementary Table 1). First we tested the acquisition of mutations in the ancestral strain of SARS-CoV2, we used in this study by performing sequencing followed by minor variant analysis. Minor variant analysis indicated the presence of five single nucleotide variants (SNVs) in ancestral strains of SARS-CoV-2 (Supplementary Table 2). None of these SNVs were found to be in the cleavage site of virus. Therefore it is highly unlikely to affect infectivity, lethality of the virus or weaken the virus's pathogenicity. This is supported by the experimental evidences we provided subsequently in which we found that infection of hACE2.Tg mice with ancestral strain of SARS-CoV2 causes lethality and mice become moribund post infection as reported earlier[24,25].

To test the role of IL-9, we infected hamster and ACE2.Tg mice with ancestral stain of SARS-CoV2. Similar to active COVID-19 patients, IL-9 was found to be upregulated in the lungs of SARS-CoV-2-infected hamster and ACE2.Tg mice (Supplementary Fig. 1b; Fig. 1b, c; Supplementary Fig. 2a, b). IL-9 is primarily produced by Th9 cells but Th2, Th17 and Foxp3+ Tregs also produce IL-9[18,26]. We tested the mRNA expression of genes, *Pu.1, Irf-1*, signal transducer and activator of transcription-5 (*Stat-5*), *Stat-6, Bat-f* that are associated with Th9 cells[9,10,13,15]. We found that genes that are associated with Th9 cells are upregulated in the lungs of SARS-CoV-2-infected hamster and ACE2.Tg mice (Supplementary Fig. 2c–e), indicating the association of Th9 cells in SARS-CoV-2 infection.

To determine the role of IL-9 in SARS-CoV-2 infection, anti-IL-9 neutralizing antibody was given to ACE2.Tg mice during SARS-CoV-2 infection at indicated time points (Fig. 1d, e). As compared to SARS-CoV-2-infection and Remdesivir (RDV) treatment, ACE2.Tg mice treated with anti-IL-9 antibody treatment resulted in rescuing weight loss and lung haemorrhage with the reduction of lung viral load (Fig. 1 f, g; Supplementary Fig. 2f, g). Similar to RDV, anti-IL-9 antibody treatment decreased immune cells infiltration, mucus production, thickening of blood vessels and mild hypertrophy and mast cell accumulation in the SARS-CoV-2-infected lungs of ACE2.Tg mice (Supplementary Fig. 2h).

Anti-IL-9 antibody treatment decreased inflammatory cytokines, IL-9, IL-4, IL-17 and IFN-γ and replenish the CD4, CD8 and γδ T cells in the BAL (Supplementary Fig. 3a–e) and expression of *Il9, Il4, Il5, Il6*, and Th9 cell-associated transcription factors, *Foxo1, Stat5, Stat6, Irf4, Bat-f, Gata3*, and *Irf1* (Supplementary Fig. 3f, g) together with allergic markers, chemokine (C-C motif) ligand 2 (*Ccl2*), CXC chemokine ligand 5 (*Cxcl5), Cxcl10*, (Tryptophan hydroxylase 1) *Tph1*, (high-affinity IgE receptor) *Fcer1* in the lung tissue samples of ACE2.Tg mice (Supplementary Fig. 3h). Contrary to IL-9 neutralization, intranasal administration of exogenous IL-9 enhanced severity of SARS-CoV-2 infection in ACE2.Tg mice as indicated by weight loss, lung lesions and viral load associated with an increased frequency of mast cells, and eosinophils in the Bronchoalveolar lavage (BAL) (Fig. 1h–l; Supplementary Fig. 1a; Supplementary Fig. 4a, b). In line with this, lung histology indicated an overall increased in immune cell infiltration, and mucus production upon IL-9 treatment in SARS-CoV-2-infected ACE2.Tg mice (Supplementary Fig. 4c). Interestingly, we found that IL-9 enhanced infection of Omicron variant of SARS-CoV-2 in ACE2.Tg mice, as indicated by loss of body weight, increased lung lesions and viral load (Fig. 1m, n; Supplementary Fig. 4d) with increased infiltration of immune cells and lung tissue damage (Supplementary Fig. 4e). We performed the minor variant analysis of Omicron stock we used in this study and identified that there are 24 SNVs. However, none of these SNVs are found in the cleavage site of the Omicron (Supplementary Table 2), indicating unlikely possibility to affect infectivity and pathogenicity of the Omicron variant is known to cause a milder infection with a reduced mortality[27].

Anti-viral pathways include IFN-stimulated genes (ISGs) and anti-viral genes, are critical for anti-viral immunity[28,29]. We found that anti-IL-9 antibody, as compared to control, treatment increased the expression of ISGs, *Irf3, Irf7, Irf9, Ifnb, Ifnar1, Ifnar2* and Interferon-induced transmembrane proteins (*Ifitm*), and anti-viral genes, adenosine deaminase acting on RNA (*Adar*), Ribonuclease L (*Rnase L*), oligoadenylate synthase2 (*Oas2), Oas3, Oas1g, Irf3, Irf7*, and *Irf9* (Fig. 1p–r). In Contrary, exogenous IL-9 inhibited the expression of anti-viral genes (*Oas2, Oas1g*, and *Oas3*) and ISGs (*Irf3, Irf7, Irf9, Ifn-α* and *Ifn-β*) in the lungs of SARS-CoV-2-infected ACE2.Tg mice (Supplementary Fig. 4f–h). In addition, we found that IL-9 treatment increased viral load in human alveolar adenocarcinoma-derived epithelial cells (Supplementary Fig. 4i). Moreover, compared to untreated ACE2.Tg mice, IL-9 treatment enhanced IL-9, IL-4, IFN-γ, and IL-13 in the BAL and IL-9R expression in the lung tissues of SARS-CoV-2 infected ACE2.Tg mice and adenocarcinomic human alveolar basal epithelial cells (A459) cell in vitro (Supplementary Fig. 5a–f).

To address the role of IL-9 in regulating the activity of the transgenic K18 promoter, which might regulate the expression of the transgenic hACE2, we performed Western Blot experiment in which we tested the expression of hACE2 at protein level in the presence or absence of IL-9 treatment. Briefly, K18hAce2 mice were administered rIL-9 (500 ng/mice) or vehicle intranasally for 24 h before euthanising the mice. Lungs from these were homogenised in protein lysis buffer for Western Blot analysis using anti-human Ace2 antibody. Our data indicated that IL-9 treatment did not increase, as compared to control treatment, hAce2 expression in K18hACE2.Tg mice (Supplementary Fig. 5g).

To further substantiate this finding, we used the intestinal epithelial cell line, Caco2, which is known to express Ace2[30,31]. Briefly, Caco2 cell line was treated with recombinant human IL-9 (10 ng/ml) for 24 h. Post IL-9 treatment, these treated cells were lysed with protein lysis buffer for further Western Blot analysis. Our results indicate that IL-9 treatment did not enhance Ace2 expression in Caco2 cells (Supplementary Fig. 5g), suggesting that IL-9 is not able to increase expression of hAce2, and thus may not influence the expression of the transgenic hACE2 in a non-physiological manner.

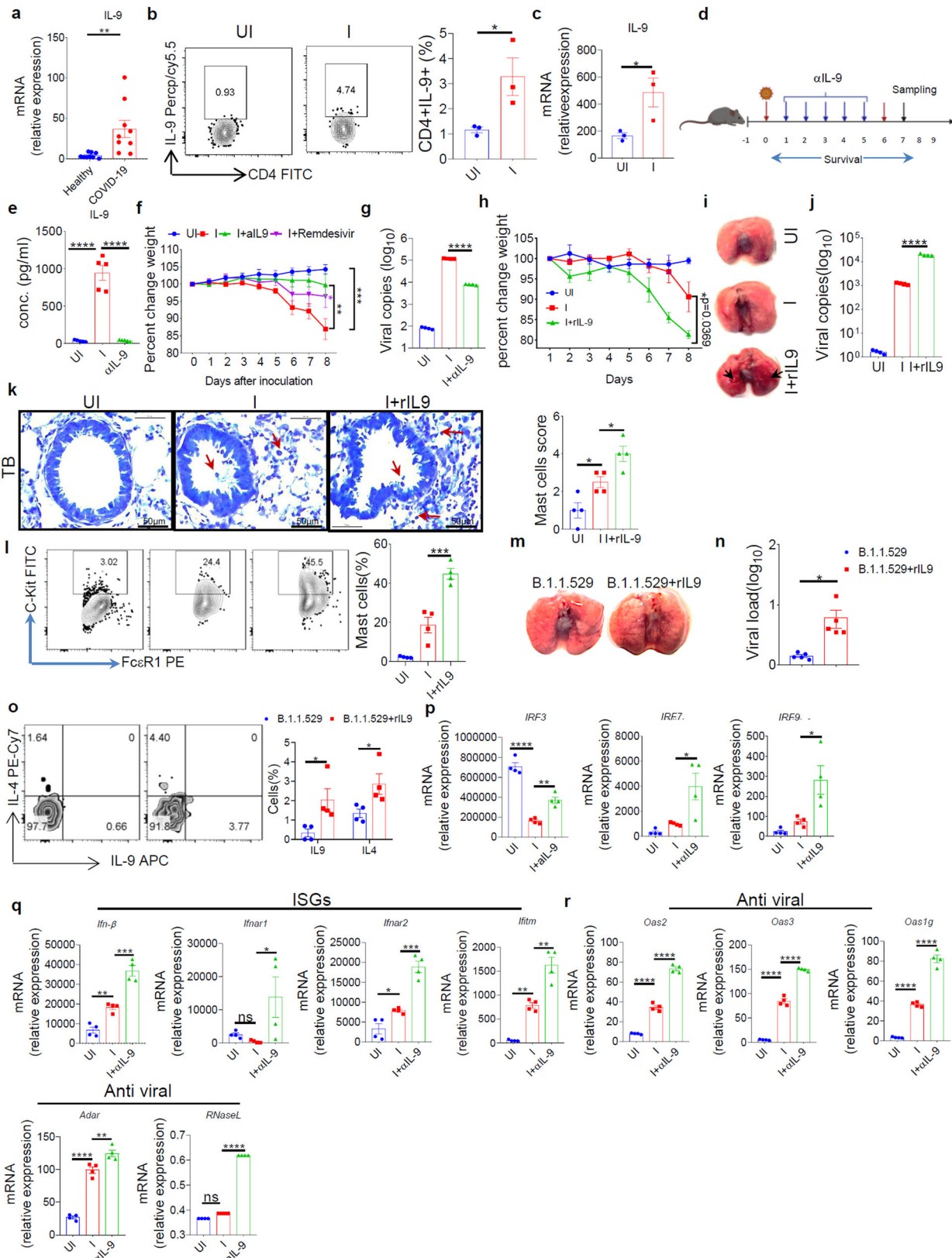

## Synergistic effect of anti-IL-9 antibody treatment with Remdesivir (RDV) in SARS-CoV-2 infection

We further tested synergistic effect of anti-IL-9 and RDV in controlling SARS-CoV-2 infection. A suboptimal (SO) dose of anti-IL-9 antibody and RDV alone or in combination was given to ACE2.Tg mice during SARS-CoV-2 infection as indicated (Fig. 2a). As compared to the optimal dose of anti-IL-9 antibody, SARS-CoV-2-infected ACE2.Tg mice treated with SO dose of anti-IL-9 antibody and RDV effectively rescued weight loss, increased survival, reduced lung lesions and haemorrhage, and decreased viral load (lung and faecal) while SO dose of anti-IL-9 antibody or RDV alone failed to do so (Fig. 2b–e). SO dose of anti-IL-9 and RDV, compared to SO of anti-IL-9 or RDV, treatment reduced the

**Fig. 1 | IL-9 promotes the severity of SARS-CoV-2 Infection and associated lung inflammation.** Role of IL-9 was studied in COVID-19 patients and in hACE2 transgenic mice by using neutralizing monoclonal IL-9 antibody. We isolated the PBMCs from Healthy (9 individuals) and COVID-19 RT-PCR positive ($n = 9$) individuals and isolated the cDNA from these PBMCs. **a** Relative mRNA expression ($2^{-\Delta\Delta Ct}$) of *Il9* in active COVID-19 patients ($n = 9$) and healthy individuals ($n = 9$), **$p < 0.01$, using student's t test; bar graph represents as a mean ± SEM. **b** Percentage frequency of CD4$^+$ IL-9$^+$ cells in BALF samples of hACE2 mice (*$p < 0.05$; students t test) bar graph represents as a mean ± SEM. **c** Relative mRNA expression of *Il9* in lung samples (*$p = 0.0428$; students t test); bar graph represents as a mean ± SEM. **d** Pictorial diagram showing therapeutic regime of αIL-9 and RDV (created with Biorender.com). **e** Quantitation of secretory Il-9 in BALF samples by ELISA (****$p < 0.0001$; one-way ANOVA followed by Tukey's Multiple comparison test); bar graph represents as a mean ± SEM. **f** Percentage change in body weight ($n = 5$; two-way ANOVA followed by Tukey's multiple comparison test. **$p = 0.0185$, ***$p = 0.0006$). **g** Relative viral load in lungs measured by qRT-PCR, ****$p < 0.0001$ (one-way ANOVA followed by Tukey's multiple comparison test). **h** Percentage change in body weight ($n = 5$). Two-way ANOVA followed by Tukey's multiple comparison test. *$p = 0.0369$. **i** Gross morphological changes of lungs ($n = 5$); black arrows represent dark red lung lesions. **J** Relative viral load was measured by qRT-PCR; ****$p < 0.0001$ (one-way ANOVA followed by Tukey's multiple comparison test); bar represents as a mean ± SEM. **k** Representative images show the Toludine blue staining and bar graph represents histological score (×60 magnifications; 50 μm scale bar) arrow represents mast cells. **l** Mast cell percentages were determined in BALF; ***$p = 0.0003$, one-way ANOVA followed by Tukey's multiple comparison test. **m** Representative image shows gross morphological changes in Omicron (B.1.1.529) infected lungs. **n** Viral load in infected (B.1.1.529) mice ($n = 5$) or mice treated with rIL-9 (500 ng/mice) measured in the lungs by qPCR (*$p = 0.0360$). **o** Representative FACS zebra plot and its corresponding bar graph showing percentage frequency of IL-9, IL-4. *$p = 0.0501$, $n = 4$. **p–r** Relative mRNA expression of transcription factors genes *Irf3, Irf7,* and *Irf9*; ISG'S genes (*Ifn-β, Ifnar1, Ifnar2, Ifitm*); and anti-viral genes (*Adar, Oas1g, Oas2, Oas3, RNaseL*) were measured in lung tissues ($n = 4$ mice per group). Bars show mean of ±SEM. *$p = 0.0201$, **$p = 0.0061$, ***$p = 0.0002$, ****$p < 0.0001$, ns = non-significant (two-way ANOVA followed by Tukey's multiple comparison test). Source data are provided as a Source data file.

frequency of eosinophils, mast cells, basophils and IL-9$^+$, IL-17$^+$ and IFN-γ$^+$ CD4$^+$ T cells in BAL and mediastinal and branchial lymph node of SARS-CoV-2-infected ACE2.Tg mice (Supplementary Fig. 1b; Fig. 2f–h). Lung histology indicated that combinatorial treatment of SO dose of anti-IL-9 and RDV reduced immune cells infiltration, mucus production and mast cell accumulation (Supplementary Fig. 6a, b), indicating that anti-IL-9 antibody and RDV treatment synergistically controlled SARS-CoV-2 infection. Altogether, we demonstrated that IL-9 regulating antiviral response while promoting airway inflammation contributes to SARS-CoV-2 infection.

## Foxo1-inhibition attenuates lung inflammation and SARS-CoV-2 infection

We previously showed that *Foxo1* is essential for IL-9 induction and Th9 cell differentiation[18]. We observed that *Foxo1* and IL-9 expression is increased in active human Covid19 patients, and in the lungs of SARS-CoV-2-infected hamster and ACE2.Tg mice (Fig. 3a–d; Supplementary Fig. 7a), suggesting that Foxo1-IL-9 axis might be critical for SARS-CoV-2 infection and lung inflammation. To test the effect of *Foxo1* in SARS-CoV-2 infection, we blocked *Foxo1* using pharmacological inhibitor (Foxo1i), AS184285, as indicated (Fig. 3e). Foxo1i inhibited *Foxo1, Pu.1* expression (Supplementary Fig. 7b) and restored weight loss with increased survival, decreased lung lesions and haemorrhage score, and decreased lung viral load of SARS-CoV-2-infected ACE2.Tg mice (Fig. 3f–h; Supplementary Fig. 7c, d). Foxo1i treatment found to be as good as RDV treatment in controlling SARS-CoV-2 infection (Fig. 3f–h). Lung histology indicated that Foxo1i as effective as RDV treatment in suppressing immune cell infiltration, collagen accumulation, mucus overproduction, and mast cell accumulation (Supplementary Fig. 7e). In line with this, Foxo1i, as compared to control, treatment reduced the frequency of eosinophils, mast cells, basophils and CD4$^+$IL-9$^+$, CD4$^+$IL-4$^+$ T cells, and IL-9 levels in BAL (Fig. 3i–n; Supplementary Fig. 7f), and suppressed Th9 cells-associated transcription factors, *IRF4 Batf*, Smad3 (Fig. 3o), in SARS-CoV-2-infected ACE2.Tg mice. Similar to IL-9 neutralization, Foxo1i suppressed the expression *Cxcl10, Ccl2, Fcer1* in the lungs of SARS-CoV2-infected ACE2.Tg mice (Supplementary Fig. 7g).

We further tested type 1 IFNs and ISGs expression to understand enhanced anti-viral activity upon *Foxo1* inhibition. We found that *Foxo1* inhibition, as compared to control, enhanced the expression of IRFs (*Irf3, Irf7, Irf9*), ISGs (*Ifitm, Ifna, Ifnb, Ifnar1, Ifnar2*), anti-viral genes (*Oas1g, Oas2, Oas3, Rnase-L, Adar, cgas* and *Sting*) (Fig. 3p–r; Supplementary Fig. 7h, i). These results suggest that *Foxo1* inhibition suppressed SARS-CoV-2 infection and airway inflammation via anti-viral pathways and IL-9, respectively. *Foxo1* exacerbates SARS-CoV-2 infection by *Foxo1*-dependent gene expression in SARS-CoV-2 infection.

To further confirm our findings that Foxo1-dependent IL-9 promotes SARS-CoV-2 infection and airway inflammation, we conditionally deleted Foxo1 in CD4$^+$ T cells in ACE2.Tg mice, and hereafter we refer these mice as Foxo1$^{fl/fl}$.CD4$^{Cre+}$ while Foxo1$^{fl/fl}$.CD4$^{Cre-}$ x ACE2.Tg mice referred as Foxo1$^{fl/fl}$.CD4$^{Cre-}$ mice. We demonstrated that Foxo1$^{fl/fl}$.CD4$^{Cre+}$, as compared to Foxo1$^{fl/fl}$.CD4$^{Cre-}$, mice were found to be less sensitive to SARS-CoV-2 infection, remained healthy with no sign of weight loss, and survived infection better than RDV treatment without substantial lung lesions and lung viral load (Supplementary video 1a, b; Fig. 4a–d; Supplementary Fig. 8a). Lung histology and immunophenotyping of BAL indicated that Foxo1$^{fl/fl}$.CD4$^{Cre+}$, as compared to Foxo1$^{fl/fl}$.CD4$^{Cre-}$, mice infected with SARS-CoV-2 showed reduced frequency of eosinophils, basophils, mast cells and IL-9$^+$, IL-4$^+$ CD4$^+$ T cells in the BAL and lower immune cell infiltration, mucus production, and mast cell accumulation in lung tissues (Fig. 4e, f; Supplementary Fig. 8b–e). Similarly, the expression of *Il4, Il5,* and *Il13,* and *Ifn-γ* Th9 cell-associated genes *Il9, Gata3, Batf* and *Pu.1* was found to be decreased in Foxo1$^{fl/fl}$.CD4$^{Cre+}$ mice, as compared to Foxo1$^{fl/fl}$.CD4$^{Cre-}$ mice (Supplementary Fig. 9a–c).

To understand the resistant phenotype of Foxo1$^{fl/fl}$.CD4$^{Cre+}$ mice to SARS-CoV-2 infection, RNA sequencing (RNAseq) was performed. Principal component analysis (PCA) indicated a distinct gene profile between uninfected, SARS-CoV-2-infected Foxo1$^{fl/fl}$.CD4$^{Cre-}$ and Foxo1$^{fl/fl}$.CD4$^{Cre+}$ mice (Supplementary Fig. 9d). Volcano plot represents differentially expressed genes between SARS-CoV-2-infected lungs of Foxo1$^{fl/fl}$.CD4$^{Cre+}$ and Foxo1$^{fl/fl}$.CD4$^{Cre-}$ mice (Supplementary Fig. 9e). Differential gene analysis indicated that, 1517, 1511 genes were upregulated and 1377, 1657 genes were downregulated in Foxo1$^{fl/fl}$.CD4$^{Cre-}$, Foxo1$^{fl/fl}$.CD4$^{Cre+}$ infected mice, respectively with respect to uninfected hACE2.Tg mice (Supplementary Fig. 9f). We selected top 100 differentially expressed genes between Foxo1$^{fl/fl}$.CD4$^{Cre-}$ and Foxo1$^{fl/fl}$.CD4$^{Cre+}$ mice, and identified the top 10 pathways, which were differentially regulated between SARS-CoV-2-infected lungs of Foxo1$^{fl/fl}$.CD4$^{Cre+}$ and Foxo1$^{fl/fl}$.CD4$^{Cre-}$ mice (Supplementary Fig. 9g). Pathways that are critical for anti-viral function including type 1 IFN, downregulation of chemokine receptor and mitogen-activated protein kinase (MAPK) activity, were differentially induced in SARS-CoV-2 infected lungs of Foxo1$^{fl/fl}$.CD4$^{Cre+}$ and Foxo1$^{fl/fl}$.CD4$^{Cre-}$ mice (Supplementary Fig. 9f), indicating that *Foxo1* controls anti-viral pathways in SARS-CoV-2 infection. Among top 100 differentially modulated genes, we found that *Daxx* (Death domain associated protein), NOD-like receptor family CARD domain containing 5 (*Nlrc5*), Serine/threonine-protein kinase PAK 2 (*Pak2*), Chemokine (C motif) ligand (*Xcl1*), *Oas2, Oasl2*, which were known to suppress SARS-CoV-2 infection[32–34], were upregulated in SARS-CoV-2-infected lungs of Foxo1$^{fl/fl}$.CD4$^{Cre+}$ mice, as compared to Foxo1$^{fl/fl}$.CD4$^{Cre-}$ mice (Fig. 4g).

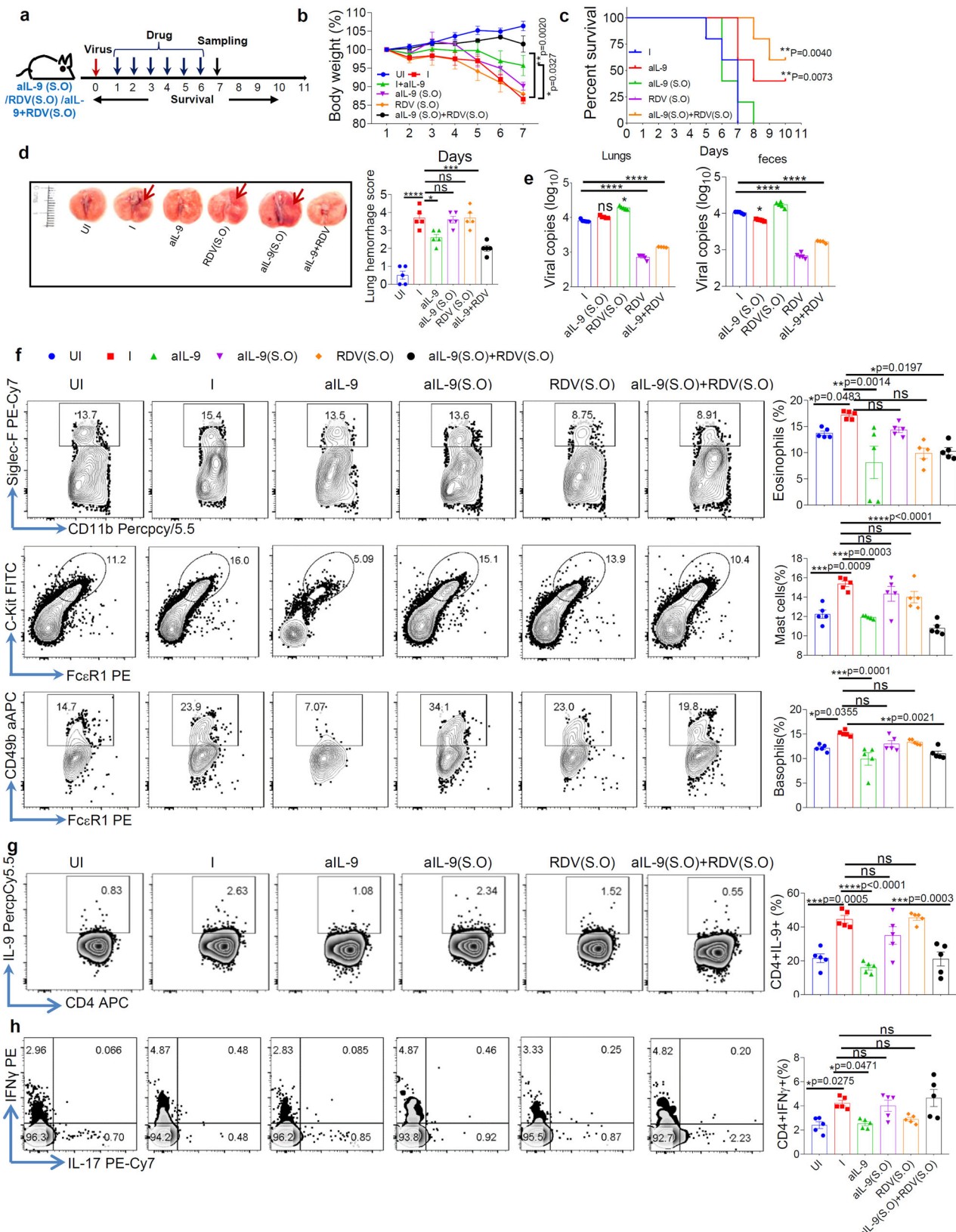

RNAseq Data identified the downregulation of *Plaa* (Phospholipase-A2-Activating protein)[35], Neural precursor cell-expressed developmentally downregulated 9 (*Nedd9)*[36], Protein Phosphatase 6 Regulatory Subunit 2 (*ppp6r2*)[37], *Slc35a3* (Human protein atlas; www.proteinatlas.org), *Serpina3M*[38] which are known to be associated with lung injury, lung cancer, allergic inflammation, fibroblast formation genes, and asthma in the lungs of SARS-CoV-2-infected lungs of Foxo1$^{fl/fl}$.CD4$^{Cre+}$ mice as compared to Foxo1$^{fl/fl}$.CD4$^{Cre-}$ (Fig. 4h–j). String analysis was performed on the differentially expressed genes between Foxo1$^{fl/fl}$.CD4$^{Cre-}$ and Foxo1$^{fl/fl}$.CD4$^{Cre+}$ mice to identify the network of genes that are either upregulated or downregulated in Foxo1$^{fl/fl}$.CD4$^{Cre+}$ mice. String analysis indicated *Nlr3c1, Irf2*, Runt-related transcription

**Fig. 2 | Synergistic effect of anti-IL-9 antibody treatment with RDV in SARS-CoV-2 infection. a** Experimental design where anti-IL-9 and RDV treatment was given as indicated (Created with BioRender.com). **b, c** Change in body weight (two-way ANOVA followed by Tukey's test) and percent survival determined by Mantel−Cox test post SARS-CoV-2 infection and treatment ($n = 5$); *$p < 0.05$, **$p < 0.005$); bar graph represents as a mean ± SEM. **d** Lung haemorrhage score at 7 dpi on a scale of 0–5, 0 is a normal pink healthy lung, and 5 is a diffusely discoloured dark red lung; bar graph represents as a mean ± SEM ($n = 5$ mice per group) (one-way ANOVA followed by Tukey's multiple comparison test) ***$p < 0.0005$, ****$p < 0.0001$, ns = non-significant; arrow represents dark red lung lesions. **e** Viral load in SARS-CoV-2-infected ACE2.Tg mice lung tissue after therapeutic treatment

($n = 5$ mice), and freshly collected faeces qPCR assays ($n = 5$ mice per group). *$p = 0.0021$, ***$p = 0.0367$, ****$p < 0.0001$; one-way ANOVA followed by Tukey's multiple comparison test; bar graph represents as a mean ± SEM. **f** Representative FACS dot plots show frequency of eosinophils, mast cells, and basophils. Bar graph represents as a mean ± SEM ($n = 5$ mice per group) (one-way ANOVA followed by Tukey's multiple comparison test). **g, h** Intracellular cytokine staining of IL-9, IFN-γ and IL-17 on CD4$^+$ T cells ($n = 5$); bar graph represents as a mean ± SEM; one-way ANOVA followed by Tukey's multiple comparison test. **g** ***$p < 0.0005$; ****$p < 0.0001$, **h** *$p < 0.05$, ns = non-significant Source data are provided as a Source data file.

factor 3 (*Runx3*), among other transcription factors were upregulated while *Smad3, Smad7*, peroxisome proliferator-activated receptor (*Ppara*) were downregulated in Foxo1$^{fl/fl}$.CD4$^{Cre+}$, as compared to Foxo1$^{fl/fl}$.CD4$^{Cre-}$, mice, and have a direct interaction with *Foxo1* (Supplementary Fig. 9i, j). While *Nlr3c1* and *Irf2* negatively regulate anti-viral functions[39,40], *Runx3* positively regulated Th1 cells and suppresses Th9 cells development[10,41] may contribute to anti-SARS-CoV-2 activity. Collectively, RNAseq Data show a distinct gene expression associated with anti-viral, immune regulation, IL-9 induction, asthma and lung injury, may make Foxo1$^{fl/fl}$.CD4$^{Cre+}$ mice resistant to SARS-CoV-2 infection. Moreover, Foxo1$^{fl/fl}$.CD4$^{Cre+}$, as compared to Foxo1$^{fl/fl}$.CD4$^{Cre-}$, mice show an increased expression of ISGs (*Ifna. Ifnb, Irf1, Irf3, Ifitm, Sting*) and anti-viral genes (*Oas1g, Oas2, Oas3, Rnase l*) in SARS-CoV-2 infected lungs (Supplementary Fig 10a, b). These Data indicate that Foxo1 promotes viral infection and associated pathology through IL-9 and suppressing anti-viral functions.

### Foxo1-IL-9 axis is essential for SARS-CoV2 transmission

Our data indicated Foxo1 linked to SARS-CoV-2 infection and regulate viral replication and shedding, raising a possibility that Foxo1 may control viral transmission. To understand this, SARS-CoV-2-infected ACE2.Tg and Foxo1$^{fl/fl}$.CD4$^{Cre+}$ mice were cohoused at 1:1 ratio with uninfected ACE2.Tg mice for 10–12 days followed by measuring various parameters of infections (Fig. 5a). As compared to uninfected ACE2.Tg mice, uninfected ACE2.Tg mice cohoused with SARS-CoV-2-infected ACE2.Tg mice develop infection while uninfected ACE2.Tg mice cohoused with SARS-CoV-2-infected Foxo1$^{fl/fl}$.CD4$^{Cre+}$ mice develop low level of infection as indicated by weight loss, lung lesions, viral load and histological score (Fig. 5b–f). In addition, we tested whether ACE2.Tg and Foxo1$^{fl/fl}$.CD4$^{Cre+}$ mice can transmit SARS-CoV-2 infection to healthy cohoused ACE2.Tg mice. Our previous data, indicate that Foxo1$^{fl/fl}$.CD4$^{Cre+}$, as compared to Foxo1$^{fl/fl}$.CD4$^{Cre-}$, mice are less sensitive to SARS-CoV-2 infection (Fig. 4a, b). This could be due to lesser viral replication in Foxo1$^{fl/fl}$.CD4$^{Cre+}$ mice. Keeping this in mind, we cohoused SARS-CoV2-Foxo1$^{fl/fl}$.CD4$^{Cre+}$ mice with healthy ACE2.Tg mice for a longer period (12-day post infection) to see whether additional period allows an efficient transmission of SARS-CoV-2 to healthy cohoused ACE2.Tg mice as it does in case of infected ACE2.Tg mice cohoused with healthy ACE2.Tg mice. Even though, longer period of cohousing of SARS-CoV-2 infected Foxo1$^{fl/fl}$.CD4$^{Cre+}$ mice were unable to transmit SARS-CoV-2 infection, indicating the inability Foxo1$^{fl/fl}$.CD4$^{Cre+}$ to transmit the infection (Supplementary Fig. 12). As compared Foxo1$^{fl/fl}$.CD4$^{Cre+}$ cohoused with ACE2.Tg, ACE2.Tg cohoused ACE2.Tg mice, showed higher frequency of eosinophils and mast cells in their BAL (Fig. 5g, h). These Data together indicated *Foxo1* limits viral transmission.

Foxo1$^{fl/fl}$.CD4$^{Cre+}$ mice were resistant to SARS-CoV-2 infection and associated mortality. Foxo1-mediated IL-9 causes severity of SARS-CoV-2 infection. We tested whether exogenous IL-9 makes Foxo1$^{fl/fl}$.CD4$^{Cre+}$ mice susceptible to SARS-CoV-2 infection. Intranasal administration of exogenous IL-9 makes Foxo1$^{fl/fl}$.CD4$^{Cre+}$ mice susceptible to SARS-CoV-2 infection as their overall physical activity declined (Supplementary video 2a, b). In line with this, exogenous IL-9 treatment of

Foxo1$^{fl/fl}$.CD4$^{Cre+}$, as compared to Foxo1$^{fl/fl}$.CD4$^{Cre+}$ mice without IL-9 treatment, mice show weight loss, lung lesions, viral load with and increased frequency of eosinophils, mast cells and IL-4$^+$-, IL-9$^+$- and IL-17$^+$-producing CD4$^+$ T cells (Fig. 5i–n; Supplementary Fig. 11a). In line with this, IL-9 treatment to Foxo1$^{fl/fl}$.CD4$^{Cre+}$ mice increased histological score as determined by various indicated changes (Fig. 5o, p). These data indicated that replenishing IL-9 in Foxo1$^{fl/fl}$.CD4$^{Cre+}$ mice makes these mice susceptible to SARS-CoV-2 infection. However, endogenous (cellular) source of IL-9 in this case was remained unclear. Since it was shown that IL-9 is primarily produced by CD4$^+$ cells and innate lymphoid cells (ILCs)[42], we tested the cellular source of IL-9 in SARS-CoV2 infection in ACE2.Tg mice. We tested IL-9 cytokine staining in CD4$^+$ T cell, NK cells and ILCs from uninfected and SARS-CoV-2 infected ACE2.Tg mice. Our data suggest the IL-9 is primarily produced by CD4$^+$ T cells; we did not find IL-9 production from NK cells and ILCs (Supplementary Fig. 13a–d).

We further tested whether CD4$^+$ T cell-driven IL-9 is sufficient in making Foxo1$^{fl/fl}$.CD4$^{Cre+}$ mice susceptible to SARS-CoV-2 infection. To do this, we transferred the wt CD4$^+$ T cells from ROSA mTmG mice into the Foxo1$^{fl/fl}$.CD4$^{Cre+}$ mice; we referred these mice as wt-CD4T-Foxo1$^{fl/fl}$.CD4$^{Cre+}$ mice. The advantage of using CD4$^+$ T cells from ROSA mTmG mice, as these CD4$^+$ T cells can be tracked in vivo based on their expression of RFP (Red fluorescence protein). Post transferring of CD4$^+$ T cells from ROSA mTmG mice, we infected wt-CD4T-Foxo1$^{fl/fl}$.CD4$^{Cre+}$ and Foxo1$^{fl/fl}$.CD4$^{Cre+}$ mice with SARS-CoV-2 ancestral strain of SARS-CoV-2. We found that wt-CD4T-Foxo1$^{fl/fl}$.CD4$^{Cre+}$ mice become as susceptible as wt ACE2.Tg mice while Foxo1$^{fl/fl}$.CD4$^{Cre+}$ mice were found to be remain resistant to SARS-CoV-2 infection (Supplementary Fig. 13e). BALF analysis suggests that IL-9 is produced from wt mTmG, but not from Foxo1-deficeint, CD4$^+$ T cells, further confirming our findings that CD4$^+$ T cell-derived IL-9 contributes to the susceptibility of SARS-CoV-2 infection (Supplementary Fig. 13f–h). Taken together, our Data demonstrate that IL-9 promote SARS-CoV-2 infection, and make in Foxo1$^{fl/fl}$.CD4$^{Cre+}$ mice susceptible to infection. Altogether, we identified one of the key mechanisms of T cell-mediated immunopathology in Covid19 by showing that *Foxo1* and IL-9 controls two distinctive pathological features that critically contribute to the progression and severity of COVID-19−namely anti-viral pathway and airway inflammation. The role of T cells was identified in mounting anti-SARS-CoV-2 response[43]. However, an Involvement of CD4$^+$ T cells was not identified in promoting airway inflammation in promoting primary SARS-COV-2 infection.

## Discussion

In this study, we identified one of the key mechanisms of T cell-mediated immunopathology in Covid19. We show that *Foxo1*-IL-9 axis controls two distinctive pathological features that critically contribute to the progression and severity of COVID-19 - namely anti-viral pathway and airway inflammation. Although the involvement of adaptive immune cells, particularly CD4$^+$ and CD8$^+$ T cells, was identified in mounting protection in primary SARS-CoV-2 infection[43], the role of CD4$^+$ T cells and its cytokines were not known in regulation of anti-viral response and airway inflammation in SARS-COV-2 infection.

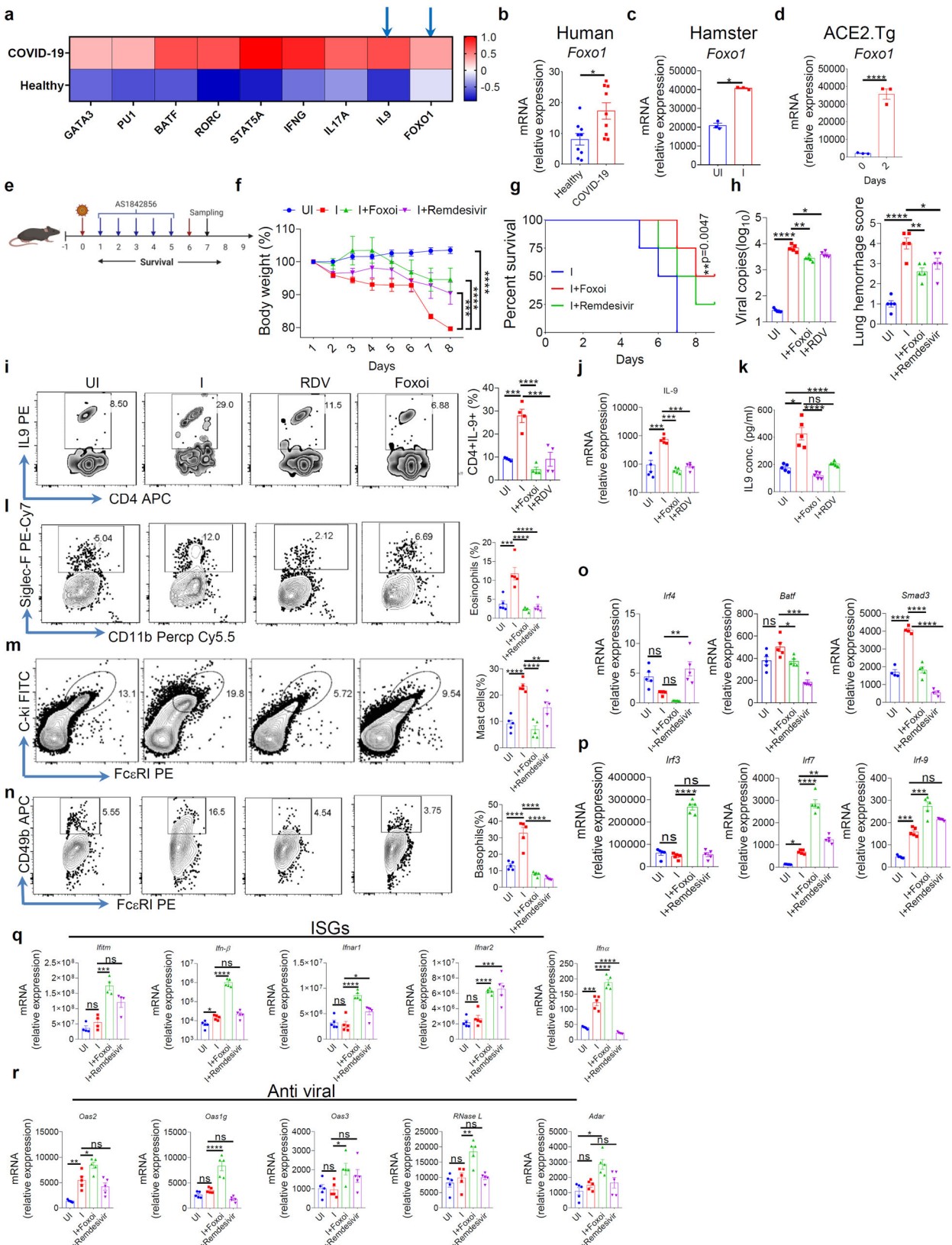

We identified an association of IL-9, a cytokine primarily produced by Th cells, in enhancing the severity of SARS-CoV-2 infection. Naïve CD4$^+$ cells differentiate into IL-9-producing Th9 cell in the presence of TGF-β and IL-4. Although the role of IL-9 and IL-9R was well documented in allergic inflammation and atopy, their functions in respiratory viral illness were not clearly understood. IL-9 was found to be present in bronchial secretion of respiratory syncytial virus (RSV) infected infants[44]. Moreover, IL-9 was found to regulate RSV-induced immunopathology and enhance RSV infection, as depletion of IL-9 promoted RSV clearances from the lungs[45]. In line with this, we show that IL-9 was found to be increased in active COVID-19 patients. Similarly, IL-9 was found to be upregulated in SARS-CoV2-infected hamster

**Fig. 3 | Foxo1 inhibition attenuates lung inflammation and SARS-CoV-2 infection. a** Heat map of gene expression of COVID-19 active patients or healthy control ($n = 9$). Gene expression levels in the heat map are z score–normalized values determined from log2 transformed relative gene expression. *Foxo1* (Human; $n = 9$) (**b**), Hamster ($n = 3$ hamster per group) (**c**), and mice ($n = 3$ mice per group) at day 2 (**d**) is represented in the form of a bar graph (*$p < 0.05$, ****$p < 0.0001$; student's t-test); bar graph represents as a mean ± SEM. **e** Pictorial diagram shows the experimental design of the study (Created with BioRender.com). **f** Percentage change in the body weight ($n = 5$ mice per group) (***$p < 0.002$, ****$p < 0.0001$; two-way ANOVA followed by Tukey's test); bar graph represents as a mean ± SEM. **g** Percent survival of each group analysed by Mantel−Cox test (**$p < 0.0047$, $n = 5$); bar represents as a mean ± SEM. **h** Relative lung viral load by qPCR assay and Lung haemorrhage scored on 7 dpi (*$p < 0.05$, **$p < 0.0047$, ****$p < 0.0001$); bar graph represents as a mean ± SEM ($n = 5$ mice per group); (one-way ANOVA followed by

Tukey's multiple comparison test). **i−k** Left panel: Intracellular staining of IL-9 was performed ($n = 4$) in BAL cells; middle panel (**j**) shows relative mRNA expression of *Il9* and right panel (**k**) shows the IL-9 concentration in BAL fluid by ELISA ($n = 5$), *$P < 0.05$, ***$P < 0.0004$, $p < 0.0001$ (right: Kruskal−Wallis test, left and middle: two-way ANOVA). **l−n** FACS plots and their respective bar graphs represent Eosinophils, Mast cells, and Basophils percentages in BALF. *$p = 0.0471$, **$p < 0.005$, ***$p = 0.0002$, ****$p < 0.0001$ (one-way ANOVA followed by Tukeys multiple comparison test) $n = 5$. **o**, **p** relative mRNA expression of *Irf4, Batf, Smad3, Irf3,Irf7,* and *Irf9* (*$p = 0.0406$, **$p = 0.0060$, ***$p = 0.0003$, ****$p < 0.0001$; one-way ANOVA). **q**, **r** relative mRNA expression of ISGs (*Ifitm, Ifnβ, Ifnar1, Ifnar2, Ifnα*) and anti-viral factors genes (*Oas2, Oas3, Oas1g, RNaseL,* and *Adar*) profile by qPCR from lung samples ($n = 4$); *$p < 0.05$, **$p < 0.005$, ***$p < 0.005$, ****$p < 0.0001$ (one-way ANOVA followed by Tukey's, multiple comparison test); bar graph represents as a mean ± SEM. Source data are provided as a Source data file.

and ACE2.Tg mice. Our data demonstrated that the depletion of IL-9 enhances viral clearance while regulates SARS-CoV-2-induced airway inflammation and lung pathology. We further characterised the role of IL-9 in SARS-CoV-2 infection and associated pathologies in ACE2.Tg mice. Our data suggested that IL-9 was found to be increased in early days of SARS-CoV-2 infection in hamster, which further indicated the involvement of IL-9 in the induction of airway inflammation. In line with this, we found that the expression of IL-9 was found to be increased in the PBMCs of active COVID-19 patients.

Role of IL-9 in inducing airway inflammation and immunopathology is well characterized, as it was shown to greatly enhance infiltration of lymphocytes, eosinophils and associated with mast cells hyperplasia and mucus production[46]. It was previously shown that anti-IL-9 neutralization markedly decreased allergic inflammation associated with reduced levels of total serum IgE, eosinophil frequency, mucus production and overall histopathological score[47,48]. In line with this, our data demonstrated that anti-IL-9 neutralization in SARS-CoV-2 infection reduced overall of histopathological score associated with the reduction in mucus production, collagen deposition and mast cell accumulation in the lungs. In line with this, exogenous treatment of IL-9 enhanced overall histopathological score in SARS-CoV-2 infection with an increased in eosinophils, mast cell accumulation and mucus production in the lungs. Taken together, our data indicated a crosstalk between IL-9 and mast cell in SARS-CoV-2 infection, which is consistent with the previous findings that IL-9 promotes mastocytosis and mast cell functions by enhancing their survival[49,50]. Although our data suggest that IL-9 enhances the severity of the SARS-CoV2 infections, mutations in the spike protein of SARS-CoV2 affect its infectivity and antigenicity. Mutation in the multi-basic cleavage site (PRRAR) at the spike protein S1/S2 interface is a major determinant of infectivity and pathogenesis of SARS-CoV2 and its variants, as it allows the efficient cleavage of S protein by the host proteases, which is a prerequisite for efficient virus entry and fusion[51,52]. The mutation of Proline (P) to Histidine (H) P681H in the cleavage site as seen in Alpha and Omicron variants has been suggested to reduce the viral infectivity as compared to the Delta virus in which the introduction of additional basic residue P681R might be responsible for the enhancement of pathogenicity[53,54]. However, except for the Proline mutation, the cleavage site is found to be highly conserved in SARS-CoV-2 variants, and mutation of the polybasic arginine residue might result in the development of low virulent strains and infection might restrict the localized spread as compared to systemic spread as shown in Influenza viruses[55,56]. In addition, mutations in other than the spike protein like D614G, N501Y, and L452R were found to be associated with viral infectivity and antigenicity. However, we did not find these mutations in the viral stocks, we used in our study.

Our data suggested that IL-9-producing Th9 cells contribute to the progression of SARS-CoV2 infection and associated airway inflammation. However, IL-9 was also found to be produced by ILCs[42],

indicating a possibility of ILCs, in addition to Th9 cells, in contributing to SARS-CoV2 infection and associated lung inflammation. We tested the role of ILCs-mediated IL-9 production in SARS-CoV-2 infection in ACE2.Tg mice. Our data indicated that neither NK cells nor ILCs contribute to IL-9 production in the lungs of SARS-CoV2 infected mice. We previously identified that *Foxo1* is one of the key transcription factors that is essential for the induction of IL-9 in Th cells[18]. Hence *Foxo1* deletion may lead to the attenuation of IL-9-mediated SARS-CoV2 infection and associated lung inflammation. In line with this, our data demonstrated that Foxo1 deficiency in CD4⁺ T cells leads to blunted IL-9 production, which makes these mice less susceptible to SARS-CoV-2 infection and associated airway inflammation. Interestingly either exogenous IL-9 or transfer of Foxo1-sufficient CD4⁺ T cells make Foxo1-deficient mice susceptible to SARS-CoV2 infection and associated inflammation further indicating the role of Foxo1 in CD4⁺ T cell-derived IL-9 in driving airway inflammation in this model.

Although IL-9-mediated regulation of anti-viral function was shown in RSV infection[45], the mechanism by which IL-9 regulates antiviral functions was not identified. It was documented that anti-viral genes and ISGs were found to be essential for the clearance of SARS-CoV-2 infection. In line with this, our data suggest that IL-9 suppresses the expression of anti-viral genes and ISGs, which could be responsible for an increased severity of SARS-CoV-2 infection in the presence of IL-9. Although there is no clear mechanism is identified as to how IL-9-meidates suppression of type 1 IFN response. Our data indicated that while anti-IL-9 antibody neutralization suppressed SARS-CoV-2 infection and increased anti-viral type 1 IFN response, exogenous recombinant IL-9 treatment enhanced SARS-CoV-2 infection and suppressed anti-viral type 1 IFN response. Upon binding to its receptor, IL-9 activates proximal signalling events primarily driven by JAK-STAT pathways, which includes *Jak1, Jak3* and *Stat1, Stat3* and *Stat5*[57]. Anti-viral response of type 1 IFNs also requires activation of STAT proteins. It was shown that *Stat3* negatively regulates type 1 IFNs response, as *Stat3* deficient cells produced enhanced type 1 IFNs, which in turn increased ISGs expression[58]. Since IL-9 activates *Stat3*, it might suppress type 1 IFNs production and function and suequently ISGs expression. This is could be one of the potential mechanisms by which IL-9 may suppress type 1 IFNs response. However, this needs to be experimentally evaluated and validated. Our data suggested that increased type 1 response together with anti-viral therapy might provide alternate therapy, as anti-IL-9 antibody with anti-viral drug, RDV, reduce both viral load and lung immunopathology. In fact, suboptimal doses of anti-IL-9 and RDV synergistically eliminate SARS-CoV-2 infection and reduce lung pathologies and inflammation. *Foxo1* is essential for the induction of IL-9 in Th9 cell, and other Th cell subsets[18]. *Foxo1* blockade reduces allergic inflammation in asthma due to reduction in IL-9 production in the lungs[18]. In line with this, our data show that *Foxo1* inhibition using pharmacological inhibitor leads to suppress SARS-CoV-2 infection and associated immunopathology. Consistently, we demonstrated that

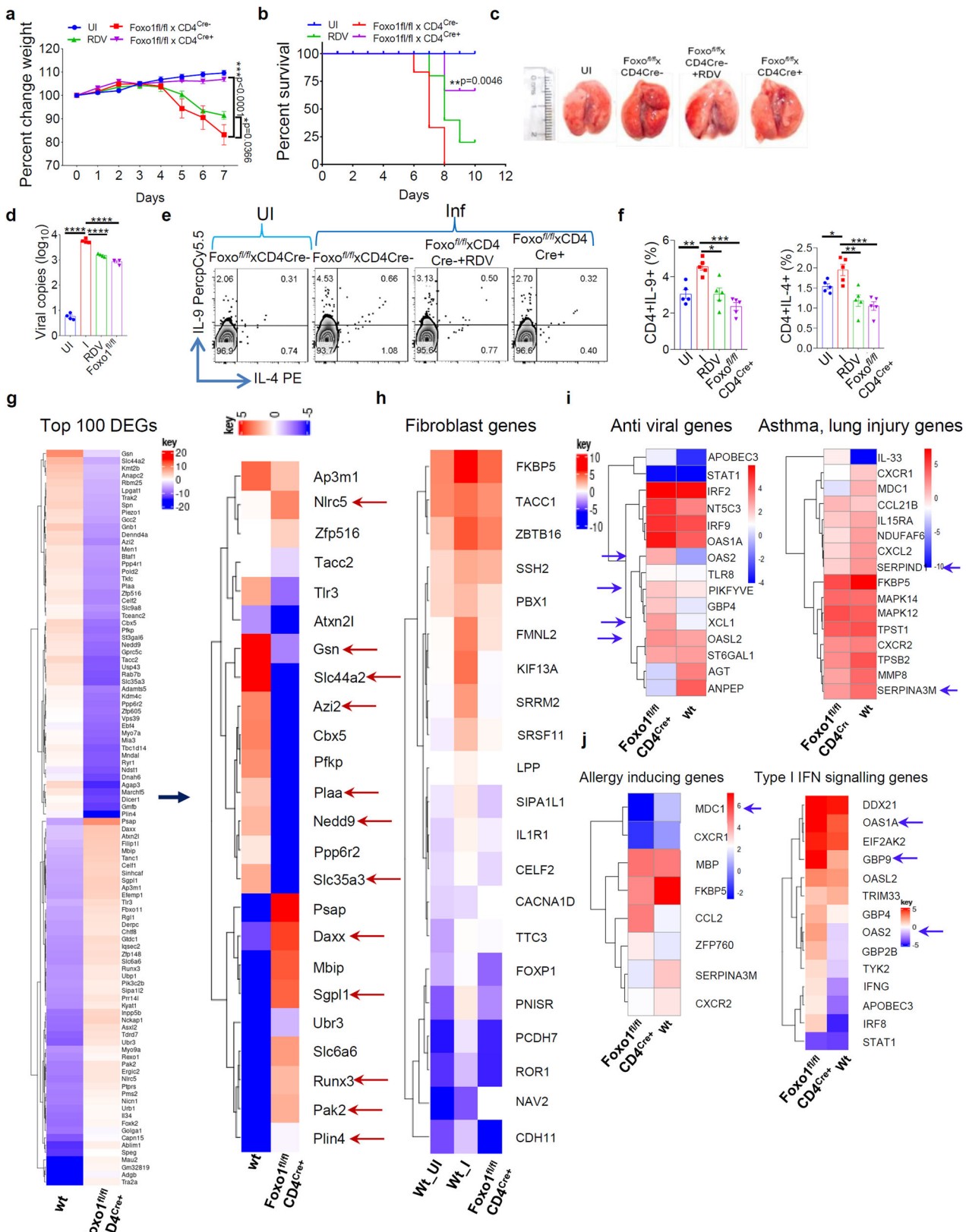

*Foxo1*-conditional deficiency in CD4[+] T cells make ACE2.Tg mice resistant to SARS-CoV-2 infection associated with the blunted IL-9 production. Mechanistically, similar to IL-9 neutralization, Foxo1 deficiency in CD4[+] T cells leads to the upregulation of anti-viral and ISGs. This could be due to the Foxo1-dependent regulation of anti-viral function by promoting *Irf3* degradation[59]. Importantly, both IL-9 neutralization and Foxo1 conditional deficiency in ACE2.Tg mice generate a similar phenotype in terms of immunopathology in SARS-CoV-2 infection associated with the accumulation of eosinophils, mast cells in the lungs and other related pathologies. Taken together, our data demonstrated that Foxo1-IL-9 axis regulates SARS-CoV-2 infection and associated immunopathology.

**Fig. 4 | Foxo1 exacerbates SARS-CoV-2 infection by Foxo1-dependent gene expression in SARS-CoV-2 infection.** hACE2.Tg (wt), Foxo$^{fl/fl}$.CD4$^{Cre+}$xhACE2.Tg (RDV group as a control) was intranasally challenged with SARS-CoV-2 and euthanization at 7 dpi. **a** Percentage changes in body Weight ($n = 5$; two-way ANOVA followed by Tukey's multiple comparison test; *$p = 0.0366$, ***$p = 0.0001$; bar graph represents as a mean ± SEM. **b** Percent survival by Mantel−Cox test (***$p = 0.0095$). **c** Image shows gross lung morphological changes **d** Relative viral load was measured by qPCR (****$p < 0.0001$); bar graph represents as a mean ± SEM ($n = 4$ mice per group); one-way ANOVA followed by Tukey's multiple comparison test. **e**, **f** BALF cells were stimulated with PMA/Ionomycin and IL-9, IL-4 production was determined by flow cytometry (*$p < 0.05$, **$p < 0.005$, ****$p < 0.0001$; $n = 5$) (one-way ANOVA followed by Tukey's multiple comparison test); bar graph represents as a mean ± SEM. **g**–**j** Heat maps of significantly up and downregulated genes during SARS-CoV-2 infection. **g** Heat map of all significantly differentially expressed top 100 genes DEGs (differentially expressed genes) and further heat-map analysis of selected top significantly differentially expressed genes. **h** Fibroblast genes **i** anti-viral & asthma, lung injury genes in SARS-CoV-2, **j** allergy, Type I IFN signalling genes identified through DEG analysis. Genes shown in each pathway are the union of the differentially expressed genes from Wt (hACE2.Tg) Vs. Foxo$^{fl/fl}$.CD4$^{Cre+}$ comparisons. Columns represent samples, and rows represent genes. Gene expression levels in the heat maps are z score−normalized values determined from log2 values. Source data are provided as a Source data file and RNA seq data available at NCBI.

## Methods

### Animals

All the experiments were performed at infectious disease research facility (IDRF) in BSL-3 and ABSL-3 as per IBSC (Institutional Biosafety Committee) guidelines. All experimental procedures involving virus challenge were approved by the Institutional Animal Ethics Committee (IAEC), IBSC and RCGM as per the guidelines of THSTI (IAEC/THSTI/191) and Department of Biotechnology, Govt. of India. Heterozygous K18-hACE2.Tg mice c57BL/6J mice (strain: 2B6.Cg-Tg(K18-ACE2)2Prlmn/J), Foxo1$^{fl/fl}$xCD4$^{Cre+}$ (Foxo$^{fl/fl}$: strain#024756; CD4Cre strain: 017336) and mTmG mice (B6.129(Cg)-Gt(ROSA)26Sortm4(ACTB-tdTomato,-EGFP)Luo/J; strain #007676) were procured from The Jackson Laboratory. Golden Syrian hamsters were procured from CDRI (Central Drug Research Institute). hACE2.Tg mice were crossed with the Foxo1$^{fl/fl}$.CD4$^{Cre+}$ mice to generate Foxo1$^{fl/fl}$.CD4$^{Cre+}$ x ACE2.Tg mice. Animals were housed and maintained at THSTI-SAF (small animal facility). All the mice were fed with a standard chow diet (Cat.no 1324p, Altromin; Germany), water and libitum. The temperature for mice rooms were maintained between -19–26 °C with ~30–70% humidity. Mice were housed with 14 h light/10 h dark cycles.

For the SARS-CoV-2 infection, mice were administered intranasally with 10³ (Foxoi, Foxo$^{fl/fl}$.CD4$^{Cre+}$ +IL-9 experiment), 10⁵ PFU (αIL-9 and other experiments), 10⁴ PFU (B.1.529 ± rIL-9) (50 µl) of live SARS-CoV-2 under injectable anaesthesia as previously described[4,60].

### Human ethics

The study was approved by the Institutional Ethics Committee (Human Research) of THSTI and ESIC Hospital, Faridabad (Letter Ref No: THS 1.8.1/(97) dated 07 July 2020). After obtaining an approval from the Institutional Ethics Committee of THSTI (IEC, Human Research) and ESIC Hospital, Faridabad active COVID-19 patients' blood samples were collected from symptomatic COVID-19 patients and healthy participants after the written informed consent and there was no bias to the recruitment or collection. PBMCs were isolated from collected blood samples, and stored in liquid nitrogen as previously described for further use[18]. Briefly, human PBMCs were isolated by Ficoll Gradient (GE Healthcare).

### SARS-CoV-2 propagation

Vero E6 (CRL-1586; American Type Culture Collection) and Caco2 (A kind gift from Dr. Sweety Samal) was cultured at 37 °C in Dulbecco's modified Eagle's medium (DMEM) 4.5 g/L D-glucose, 100,000 U/L Penicillin471 Streptomycin, 100 mg/L sodium pyruvate, 25 mM HEPES and 2% FBS The isolate of SARS-CoV-2; USA-WA1/2020 and B.1.1.529 (10⁴ PFU) was used as a challenge strain As mentioned above[53,60] at THSTI Infectious Disease Research Facility (Biosafety level 3 facility).

### In vivo treatments of RDV, anti-IL-9, Foxoi and exogenous IL-9

All the studies were designed to study therapeutic efficacy as compared to RDV. Therapeutic studies (Neutralisation experiments) were performed to define if drug regimens could affect virus replication and disease progression. 6–8-week-old mice (Male and Female) were used

in this experiment. All treatments were initiated one day prior to infection. Post challenge, uninfected and infected mice were observed till the infected mice lost 20-30% weight and the animals were sacrificed when they became moribund for further validation. In two other experiments, treatment with a vehicle, RDV (25 mg/kg; ip), anti-IL-9 (20 µg/mice; ip), and Foxo1 inhibitor (20 mg/Kg; i.n.) were given. On appearance of moribund features, animals were euthanized by isoflurane; lungs were scored for haemorrhage (described below). The left lobe was placed in 10% formalin and stored at room temperature until sectioning and histological analysis. Lung sectioning, haematoxylin, eosin staining, and N antigen staining as described below was performed at the Institute of Liver and Biliary Sciences (ILBS) at New Delhi.

We then performed two therapeutic studies to ascertain whether αIL-9 (suboptimal, S.O), RDV (S.O), αIL-9(S.O) + RDV (S.O), and Foxo1 inhibitor alone could affect virus replication or disease progression. In the first study, in randomly allocated groups ($n = 5$; 6–8- week-old mice (male and female)), we compared vehicle, RDV, or Foxo1 inhibitor alone. In another study, S.O of αIL-9 (10 mg/kg) and RDV (S.O) (1.5 mg/kg), aIL-9 (S.O) + RDV (S.O) (10 mg/kg + 1.5 mg/kg respectively) were administered once daily via intraperitoneal injection.

To evaluate the role of IL-9, we used Foxo1$^{fl/fl}$.CD4$^{Cre+}$ mice compared to hACE2.Tg mice infection with rIL-9 i.n. (i.n.; 500 ng/mice; 6–8-week-old mice (male and female) were used in this experiment.). Euthanasia and immunopathological assays were performed as described above.

### Cohousing and viral transmission

hACE2.Tg mice (6–8 weeks male and female) and Foxo1$^{fl/fl}$.CD4$^{Cre+}$ACE2.Tg (6–8 weeks male and female) mice were infected with the with SARS-CoV-2. 24 h post infection, infected mice were cohoused with the uninfected ACE2.Tg mice (1:1 ratio). Subsequently, mice were followed for the sign of infection and other parameters were measured.

### Measurement of viral load

Lung tissues and faecal samples were weighed and homogenized for further processing as described earlier[60]. Briefly, RNA was extracted by using Total RNA Isolation Kit (MDI) as per the manufacturer's protocol. Relative copy number of SARS-CoV-2 RNA was done using previously used formula (POWER (2, −ΔCT)*10,000 to calculate the relative gene expression[60]. cDNA was synthesized using the kit from applied biosystem. Copies of SARS-CoV-2 nucleocapsid (N) RNA were determined using the N gene primers (forward: 5′-ATGCTGCAATCGTGCTACAA-3′; reverse: 3′-GACTGCCGCCTCTGCTC-5′. β-actin gene was used as an endogenous control for normalization. ΔΔCt method was used for relative quantitation[18,61].

### Peripheral blood mononuclear cells (PBMCs) isolation

PBMCs were isolated using density-gradient centrifugation as described previously[18]. Briefly, human PBMCs from healthy donors were isolated by Ficoll-paque (GE Healthcare) gradient, and the PBMCs were then washed once with 1X PBS, followed by isolation of total RNA using

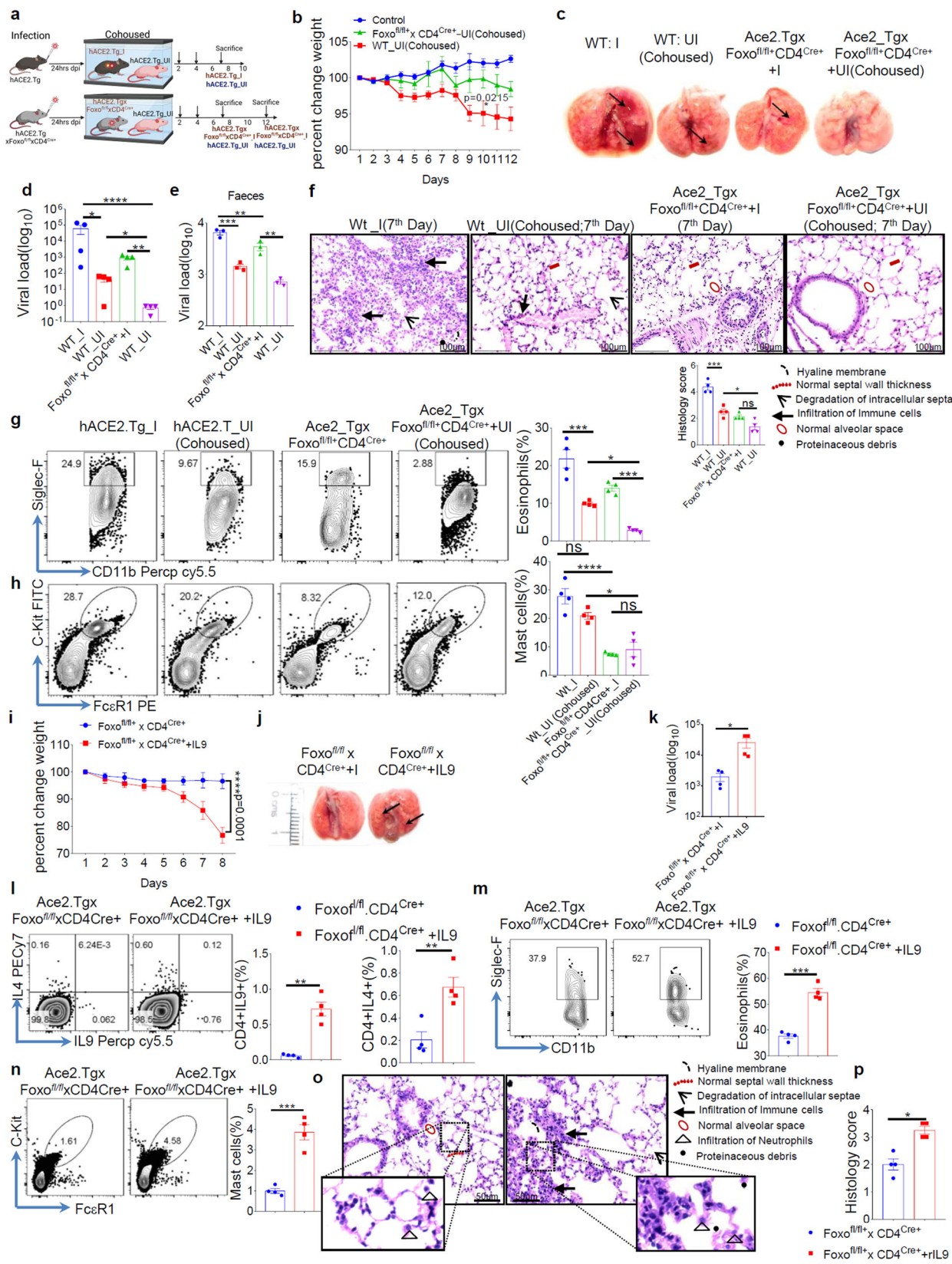

Trizol reagent. The RNA isolated from PBMCs was used to test the expression respective genes relative to β-Actin.

### Gene expression profiling of human PBMCs
Peripheral blood mononuclear cells (PBMCs) were isolated using density-gradient centrifugation as described previously[62]. Briefly,

blood samples were collected from symptomatic COVID-19 patients (~0–3 days from PCR positive report) and healthy volunteers in heparinized CPTTM (BD Biosciences, USA) tubes and were centrifuged at 1500 × g for 25 min. The PBMC layer was separated and washed with 1X PBS, and total RNA from PBMCs was isolated using the Trizol reagent. The RNA isolated from PBMCs was used to test the expression

**Fig. 5 | Foxo1-IL-9 axis is essential for SARS-CoV2 transmission.** hACE2.Tg mice (uninfected) were cohoused with Foxo$^{fl/fl}$.CD4$^{Cre+}$ infected (10$^3$PFU) mice, while hACE2-Tg healthy mice were cohoused with hACE2-Tg infected mice. **a** Pictorial diagram represents cohousing of mice (Adopted from "cohousing template" by BioRender.com 2020). **b** Percentage body weight change ($n = 5$; two-way ANOVA: *$p = 0.0215$); bar graph represents as a mean ± SEM. **c** Gross pathology of Lungs (black arrows show focal red lung lesions). **d, e** Viral burden in the lungs and faeces was analysed by qPCR for viral RNA levels ($n = 4$; two-way ANOVA followed by Tukey's multiple comparison test (*$p < 0.05$, **$p < 0.005$, ***$p < 0.0005$, ****$p < 0.0001$); bar graph represents as a mean ± SEM. **f** Haematoxylin-Eosin staining of lung sections; bar graph represents histology score (Hyaline membrane, septal wall thickness, degradation of intracellular septae, infiltration of neutrophils and protein debris); (×40 magnification; 100 μm scale bar); $n = 5$ mice per group (two-way ANOVA followed by Tukey's multiple comparison test); *$p < 0.05$, **$p < 0.005$. FACS representative dot plot and its respective bar graph show percentage frequency mean ± SEM indicating **g** Eosinophils (upper) ($n = 4$), **h** Mast cells (Lower) percentage frequency ($n = 4$); bar graph represents as a mean ± SEM;

*$p < 0.05$, ***$p < 0.0005$, ****$p < 0.0001$, ns = non-significant. Further to check the role of IL-9 in Foxo1$^{fl/fl}$ CD4Cre $^+$ mice we have given exogenously IL-9 to the mice and observed. **i** Percentage body weight change was monitored (one experiment; $n = 5$; two-way ANOVA *$p < 0.0451$; bar graph represents as a mean ± SEM. **j** Gross lung morphological changes between Foxo$^{fl/fl}$. CD4$^{Cre+}$infected and rIL-9 treated groups. **k** Viral load quantified by qPCR ($n = 4$); *$p < 0.05$ (students t test); bar graph represents as a mean ± SEM. **l–n** Representative FACS dot plot and its corresponding bar graph showing mean ± SEM showing IL-9, IL-4, Mast cells and Eosinophils frequency ($n = 5$ mice per group); one-way ANOVA followed by Tukey's multiple comparison test (*$p < 0.05$, **$p < 0.005$, ***$p < 0.0005$). **o, p** Histopathology of lungs from SARS-CoV-2 infected Foxo1$^{fl/fl}$.CD4$^{Cre+}$ mice and rIL-9 treated mice at day 7 post infection. Histopathological observations were done for the H&E stained sections and histological score was given based on Hyaline membrane, septal wall thickness, degradation of intracellular septae, infiltration of neutrophils and protein debris (×60 magnification; 50 μm scale bar). $n = 4$; bar graph represents as a mean ± SEM; *$p < 0.05$ (student t test). Source data are provided as a Source data file.

of respective genes relative to β-Actin by RT-PCR. The relative expression levels (2$^{-\Delta\Delta Ct}$) of genes were further normalized by log2 transformation, and Z-score was calculated as described previously[63]. The median of Z-scores of log2 transformed relative gene expression was represented as a heat map.

## qPCR

RNA was isolated from lungs and spleen as described previously[18]. RNA from BAL cell samples was isolated by using RNAeasy kit (MDI). Total RNA was subjected to cDNA synthesis using the iScript cDNA synthesis kit (Biorad; #1708891). qPCR was performed as described earlier using SYBR green dye KAPA SYBR FAST qPCR Master Mix (2X) Universal kit on standard 7500 Dx real-time PCR system (Applied Biosystems). The relative gene expression was calculated as described previously[18]. Following primer sets were used. Mice Primer sets: *Il-13*−5′-CTTAAG-GAGCTTATTGAGGAG-3′ 3′-CATTGCAATTGGAGATGTTG-5′; *cGAS*−5′-GGATAGAGAAAACATGCTGTG-3′; 3′-CAGTTTTCACATGGTAGGAAC-5′; *Tmem173*−5′-CTCATTGTCTACCAAGAACC-3′ 3′-TAACCTCCTCCTT TTCTTCC-5′; *Tbk1*−5′-GAACAACTCAATACCGTAGG-3′ 3′-AATTCTT-GATAGAGCAGCAG-5′; *Irf3*−5′-CTTGTAGAATAACCACCAGC-3′ 3′-CTT GTAGAATAACCACCAGC-5′; *Cxcl10*−5′-AAAAAGGTCTAAAAGGGCTC-3′ 3′-AATTAGGACTAGCCATCCAC-5′; *Cxcl5*−5′-TCAGAAAATATTGGGC AGTG-3′ 3′-CAAAGCAGGGAGTTCATAAAG-5′; *Oas1g*−5′-CTGTGGTAC CCATGTTTTATG-3′ 3′-ATACATGTCCAGTTCTCCTC-5′; *Oas2*−5′-TTA-TAAAAATACCGGCAGCTC-3′ 3′-ATTACAGGCCTCTTTTTCTG-5′; *Oas3*−5′-CCAAACTTAAGAGCCTGATG-3′ 3′-GCCTCTCCTCCTTTATATCG-5′; *RNasel*−5′-ATACTGTAGGTGATCTGCTG-3′ 3′-AAGTATCTCCTTCATTC CCC-5′; *Ifnβ1*−5′-AACTTCCAAAACTGAAGACC-3′ 3′-AACTCTGTTTTC CTTTGACC-5′; *Ifnar2*−5′-AGCCCAAAGTGAATAATGTC-3′ 3′-TGA-TAATCCTGATTCCTGGC-5′; *Ifnar1*−5′-CTGAATAAGACCAGCAACTTC-3′ 3′-CATGACAGAGAAGAACACAAC-5′; *Il-5*−5′-CCCTACTCATAAAAT CACCAG-3′ 3′-TTGGAATAGCATTTCCACAG-5′; *Ifitm3*−5′-AAGAATCAA GGAAGAATATGAGG-3′ 3′-GATCCCTAGACTTCACGG-5′; *Trim24*−5′-TTCCATCTCTCATCAGCATC-3′ 3′-CATTCTGGCTTGGTGAATATC-5′; *β-Act*−5′-TTAATTTCTGAATGGCCCAG-3′ 3′-GACCAAAGCCTTCATACA TC-5′; *Foxo1*−5′-AAACACATATTGAGCCACTG-3′ 3′TCTACTCTGTTT-GAAGGAGG5′; *Ccl2*−5′-GAAGATGATCCCAATGAGTAG-3′ 3′-TTGGTGA CAAAAACTACAGC-5′; *Ccl12*−5′-TGTGATCTTCAGGACCATAC-3′ 3′-CATGAAGGTTCAAGGATGAAG-5′; *Tph1*−5′-GAACTCAAACATGCACTT TC-3′ 3′-GTTGTACTTCAGTCCAAACG-5′; *Fcεr1a*−5′-TCAACTACAGTTA TGAGAGCC-3′ 3′-TGGGAAAATTAGTTGTAGCC-5′; *Il-17*−5′-ACGTTTCT CAGCAAACTTAC-3′ 3′-CCCCTTTACACCTTCTTTTC-5′; *N1*−5′-GACCC-CAAAATCAGCGAAAT-3′ 3′-TCTGGTTACTGCCAGTTGAATCTG-5′ *Il-9*−5′-GCATCAGAGACACCAATTAC-3′ 3′-GTACAATCATCAGTTGGGAC-5′; *Ifn-γ*−5′-TGAGTATTGCCAAGTTTGAG-3′ 3′-CTTATTGGGACAATCTCTT CC-5′; *Irf-4*−5′-GAGTAGGATCTACTGGGATG-3′ 3′-CTTGCAGCTCTGA

TAGAAAC-5′; *Irf9*−5′-CAACATAGGCGGTGGTGGCAAT-3′ 3′-GTTGATG CTCCAGGAACACTGG-5′; *Irf-7*−5′-CCACGCTATACCATCTACCTGG-3′ 3′-GCTGCTATCCAGGGAAGACACA-5′; *hACE2*−5′-TCCATTGGTCTTC TGTCACCCG-3′ 3′-AGACCATCCACCTCCACTTCTC-5′; *Spi1*−5′-GAGG TGTCTGATGGAGAAGCTG-3′ 3′-ACCCACCAGATGCTGTCCTTCA-5′.

Human primer sets: *Oas2*−5′-GCTTCCGACAATCAACAGCCAAG-3′ 3′-CTTGACGATTTTGTGCCGCTCG-5′; *Il9r*−5′-GACCAGTTGTCTCTGTT TGGGC-3′ 3′-TTTCACCCGACTGAAAATCAGTGG-5′; *Ifna*−5′ TGGGCTG TGATCTGCCTCAAAC-3′ 3′-CAGCCTTTTGGAACTGGTTGCC-5′; *Irf9*−5′-CCACCGAAGTTCCAGGTAACAC-3′ 3′-AGTCTGCTCCAGCAAGTATCG G-5′; *Oas1*−5′-AGGAAAGGTGCTTCCGAGGTAG-3′ 3′-GGACTGAGGAAG ACAACCAGGT-5′; *Oas3*−5′-CCTGATTCTGCTGGTGAAGCAC-3′ 3′-TCC CAGGCAAAGATGGTGAGGA-3′; *Ifnβ*−5′-CTTGGATTCCTACAAAGAAG CAGC-3′ 3′-TCCTCCTTCTGGAACTGCTGCA-5′; *Trim22*−5′-GGATCGTC AGTAGAGATGCTGC-3′ 3′-GAACTTGCAGCATCCCACTCAG-5′; *Ifitm*−5′-GGCTTCATAGCATTCGCCTACTC3′ 3′-AGATGTTCAGGCACTTGGCGG T-5′; *RNaseL*−5′-AAGGCTGTTCAAGAACTACACTTG-3′ 3′-TGGATCTC CAGCCCACTTGATG-5′; *Il9*−5′-GACATCAACTTCCTCATC-3′, 5′-GAGAC AACTGGTCTTCTGG-3′; *Irf3*− 5′-TCTGCCCTCAACCGCAAAGAAG-3′ 3′-TACTGCCTCCACCATTGGTGTC-3′; *Irf7*−5′-CCACGCTATACCATCTAC CTGG-3′ 3′ -GCTGCTATCCAGGGAAGACACA-5′; *Il9r*−5′-ATCAGTCCT GCCTTGGAGCCAA-3′ 3′-CCGACAATGTGATCCCTGTGCT-5′; *Ifn-α*−5′-T GGGCTGTGATCTGCCTCAAAC-3′ 3′-CAGCCTTTTGGAACTGGTTGC C-5′; *Ifn-β*−5′-CTTGGATTCCTACAAAGAAGCAGC-3′ 3′-TCCTCCTTCTG GAACTGCTGCA-5′; Hamster primer sets: *Foxo1*5′-AGGATAAGGGCGA-CAGCAAC-3′ 3′-GTCCCCGGCTCTTAGCAAAT-5′; *Il-9*−5′-CTCTGCCCTG CTCTTTGGTT-3′ 3′-CGAGGGTGGGTCATTCTTCA-5′; *Pu.1*−5′-GCATTG GAGGTGTCTGAT-3′ 3′-CATCTTCTTGCGGTTGCCCT-5′.

## BALF collection and lung histological analysis

Lung lavage was collected by inserting a cannula into the trachea and lavaging with 500 μl cold PBS three times as previously described[12]. Lavage sample was centrifuged, and supernatant was collected for further cytokine analysis, and cells were used for FACS analysis. Lungs were excised; left lower lobe was immersed in 10% formalin and used for histological analysis. Paraffin-embedded tissue samples were further sectioned and various histological staining's (H&E, Periodic acid Schiff's, van Gieson, and toluidine blue) were performed at ILBS (New Delhi) and the scoring was done by a trained histopathologist independently in a blinded manner.

Lung Injury Scoring System given by the three random people, in order to help quantitate histological features of ALI (Acute Lung Injury). In a blinded manner, three random diseased fields of lung tissue were chosen at high power (60×), which were scored for the following: (A) Immune cell infiltration (none = 0, 1–5 cells = 1, >5 cells = 2), (B) Damage in the interstitial space/septae, (C) Proteinaceous

debris in air spaces (none = 0, one instance = 1, >1 instance = 2), (D) alveolar septal thickening (<2× mock thickness = 0, 2–4× mock thickness = 1, >4× mock thickness = 2).

## Immunohistochemistry

For immunohistochemistry 2–4-μm sections were used as described earlier[12]. Briefly, paraffin-embedded sections were dewaxed and rehydrated through xylene and graded alcohol, respectively, for 15 min at room temperature (RT), before epitope unmasking, slides were blocked with normal goat serum for 30 min in RT. Samples were then incubated with a primary antibody incubated overnight at 4 °C (SARS-CoV-2-N antigen (5–25 μg/ml; R&D, # Clone # 1035145)). Species-matched gamma globulin was used as an isotype control at the same concentration. Sections were washed in PBST and Species-matched secondary antibodies were applied for 60 min at RT. The finally stained sections for SARS-CoV-2-N protein were then observed and images were captured under Ti Eclipse Nikon microscope at RCB (Regional Centre for Biotechnology).

## Cytokine ELISA

Quantitation of IL-4, IL-9, IL-10 and IFN-γ was measured in BALF ELISA as described earlier[12]. Briefly, ELISA plates were coated overnight by anti-IL-4, anti-IL-9, anti-IL10 or anti-IFNy antibodies overnight in bicarbonate buffer. Thereafter, wells were washed and blocked and then incubated with BAL fluid at 1:1 dilution. The wells were then washed and incubated with detection antibody conjugated with biotin. Colour was developed by incubating with Avidin-HRP enzyme and then with TMB substrate. Reaction was stopped by using 0.2 N stop solution and plate was read at 600 nm in spectrophotometer (BioLinkk).

## Flow cytometry analysis

FACS analysis of BAL cells, Splenocytes, and dLN's were carried out using fluorochrome labelled antibodies. Cells were collected from the lungs after washing them with the cold PBS. Cells were activated with PMA (phorbol 12-myristate13-aceate; 50 ng ml⁻¹; Sigma-Aldrich) and Ionomycin (1.0 μg ml⁻¹; Sigma-Aldrich) in the presence of Monensin (#554724 Golgi Stop, BD Biosciences) followed by surface markers and intracellular cytokines staining. The following antibodies were used: anti-mouse CD3 BV510 (#100353, Clone-145-2C11, Biolegend INC, USA, 1:1000), anti-mouse γδTCR FITC (#118105, Clone-GL3, Biolegend INC, USA, 1:1000), anti-mouse Gr1 BV421 (#108445, Clone-RB6-8C5, Biolegend INC, USA, 1:1000), anti-mouse CD11b PerCp-Cy5.5 (#101228, Biolegend INC, USA, 1:1000), anti-mouse CD4-Percp cy5.5 (#100538; Clone-RM4-5, Biolegend INC, USA, 1:1000), anti-mouse CD4-FITC (#100406; Clone-GK1.5, Biolegend, INC, USA, 1:1000), anti-mouse NK1.1–PE-Cy7 (#108714; Biolegend, USA, 1:1000), anti-mouse-CD8 – BV421 (#100753; Clone-53-6.7 Biolegend INC, USA,2:2000), F4/80 – FITC (#123108; Clone-BM8, Biolegend INC, USA, 1:1000), CD206 – PE (#141705; Clone-C068C2, Biolegend, USA, 2:1000), CD80 – AF647 (#305216, Clone-2D10, Biolegend INC, USA, 1:1000), CD68 – PEcy7 (#137015; Clone-FA-11, Biolegend INC, USA, 1:1000), CD49b (#117322; Clone-N418, Biolegend INC, USA, 2:100), C-kit (#105805; Clone-2B8, Biolegend INC, USA, 2:100), Fcer1 (#134308; Clone-MAR1, Biolegend INC, USA, 2:100), Siglec-f (#155528; Clone-S17007L, Biolegend INC, USA, 2:100), IFNγ – AF647 (#505814; Clone-XMG1.2, Biolegend INC, USA, 1:100), IL-17 – PE-cy7 (#506922; Clone-TC11-18H10.1, Biolegend INC, USA, 1:100), IL-10 – PE (#505008; Clone-JES5-16E3, Biolegend INC, USA, 5:1000), Foxp3 – AF647 (#126408; Clone-MF14, Biolegend, USA, 2:500), IL-9 – Percp-cy5.5 (#514112, Clone-RM9A4, Biolegend INC, USA, 5:1000), IL-4 – PE (#504104, Clone-11B11, Biolegend, USA, 2:500) IL-4 – PE-Cy7 (#504118, Clone-11B11, Biolegend, USA, 1:200).

NK, ILCs analysis of BALF cells were carried out using fluorochrome labelled antibodies. Cells were activated as mentioned above followed by surface markers and intracellular cytokines staining. The following antibodies were used: Lineage cocktail antibodies {anti-mouse CD3 FITC (# 100204, Clone-17A2, Biolegend INC, USA, 2:2000), anti-mouse CD11b FITC (#101206, Clone-M1/70, Biolegend INC, USA, 1:1000), F4/80 – FITC (#123108; Clone-BM8, Biolegend INC, USA, 1:1000), anti-mouse B220- FITC (#103206; Clone-RA3-6B2, Biolegend INC, USA, 2:2000), anti-mouse CD4-Percp cy5.5 (#100538; Clone-RM4-5, Biolegend INC, USA, 1:1000), anti-mouse NK1.1–PE-Cy7 (#108714; Clone-PK136, Biolegend INC, USA, 1:1000), IL-9 – APC (#514106; Clone-RM9A4, Biolegend, USA, 3:3000). Stained cells were acquired on FACS-Canto-II (Becton Dickinson, San Jose, CA) and analysed using Flowjo software (Tree Star, Ashland, OR, USA).

For sorting of mTmG CD4⁺, Foxo^fl/fl.CD4^Cre+ CD4⁺ T cells, we sacrificed the ROSA mTmG wt mice and Foxo^fl/fl.CD4^Cre+ mice euthanized; spleen and lymph nodes were collected aseptically and single-cell suspensions were made from 6–8-week-old mice. CD4⁺ T cells were purified using anti-mouse CD4-Percp cy5.5 (#100538; Biolegend). CD4⁺ T cells were further sorted using fluorescence-activated cell sorting (FACS) on BD FACSAriaIII (BD Biosciences) to obtain PE⁺ CD4⁺ T cells and CD4⁺ T cells using anti-CD4-Percp Cy5.5. The purity of sorted cells was typically ~96% in post-sort analysis.

## hACE2 expressing Intestinal epithelial cells isolation (IECs)

Primary IECs were isolated as described earlier[64]. Briefly, the intestine from the ACE2.Tg mice was cleaned with the cold PBS containing Gentamycin The intestine was cut longitudinally and mucus layer was removed and placed in EDTA (30 mM) solution at 4 °C for 20 min. Colon epithelial cells were gently removed and washed with PBS. Subsequently, the collected cells were plated in collagen-coated plates.

## Western blot

We treated the hACE2.Tg mice with or without rIL-9 (500 ng/mice; i.n.). After 24 h, we sacrificed the mice using an overdose of Ketamine and Xylazine. Cardiac perfusion was performed, and further, we digested the lungs with Dispase to get the single-cell suspension. We collected the cells and lysed them in RIPA buffer containing PIC (Protease inhibitor cocktail). In line with this, we treated the Caco2 cells (Human epithelial cell line) with or without rIL-9 (10 ng/ml) for 24 h. We have lysed the cells in RIPA buffer, as mentioned above. The protein concentration was determined by performing a BCA protein assay (Bio-Rad). 40 μg of protein extracts were loaded on a 10% SDS-Gel for hACE2. Proteins were transferred to a membrane, blocked with 5% BSA, and then incubated in primary antibody (MA5-31395; Thermo-Scientific) overnight at 4 °C. The HRP-conjugated anti-mouse (#7076; CST) secondary antibody was incubated for 1 h. bands were captured on Gel-doc (BIO-RAD). Band intensities were normalised with β-Actin (#4967; CST) and calculated by using ImageJ software.

## Transcriptome profiling using RNA quantification sequencing

RNA sequencing of SARS-CoV-2 infected lungs of ACE2.Tg, hACE2.Tgx Foxo^[.CD4^Cre-, and Foxo^fl/fl.CD4^Cre+ SARS-CoV-2 infected lung tissues were homogenised and RNA was derived and subjected to next-generation sequencing (NGS) to generate deep coverage RNASeq data. Size selection of RNA fragments was done with SPRI Beads-based Size Selection. High-quality libraries were prepared using NEB Next Ultra II Directional RNA Library Prep Kit according to manufacturer's protocols and paired-end reads of 151 bp read length were generated on the Illumina Novoseq 6000 platform.

## Transcriptome analysis

Quality-based filtering and adaptor trimming of the raw sequencing reads was done using fastp (v0.20.1). A thresh hold of 30 was set for the phred quality score. The filtered reads were aligned against the Mus musculus (mm39) genome using the splice aware aligner Hisat (v2.2.1).

The alignments were assembled into transcripts with stringtie assembler (v2.1.5). Stringtie computes read counts for the genes and

normalized expression values with the Transcript per million (TPM) metric. The gene read counts were used for differential analysis between the conditions. Genes having a p-value of less than 0.05 were considered to have a significant differential change in the expression between the conditions. A log2Foldchange of 2 and higher of these significant genes were classified as upregulated and a log2Foldchange of −2 and lower as downregulated. Genes were functionally annotated with Gene Ontology terms and Reactome pathways using NCBI resources. David Bioinformatics resources (v6.8) was used to identify significant enrichment of significant GO terms and pathways. String database was used to determine for interaction of the protein-coding genes. A high confidence score of 0.9 was used to compute the interactions. R packages used for visualization – Complex Heat map, Enhanced Volcano, ggplot2.

### Principal component analysis
High-dimensional expression data are mathematically reduced to principle components that can be used to describe variation across samples in fewer dimensions to allow human interpretation. Principle component 1 (PC1) accounts for the most amount of variation across samples, PC2 the second most, and so on. These PC1 vs PC2 plots are coloured by sample annotation to demonstrate how samples cluster together (or not) in reduced dimensional space. PCA emphasizes variation and brings out strong patterns in a dataset. It's used to make data simpler to explore and visualize.

### Viral RNA isolation and qRT-PCR
Viral RNA from stock solutions was isolated using Trizol, and SARS-CoV-2 detection and quantification were performed using a SARS-CoV-2 kit (Illumina, Cat. No. 20044311) with a cycle threshold of 35.

### Library preparation, sequencing
Whole genome sequencing of the SARS-CoV-2 and B.1.1.529 samples, using the capture-based Illumina Respiratory Virus Oligo Panel (RVOP), was done to capture the SARS-CoV-2 genome. The library preparation protocols for RVOP have been previously described. Briefly, double-stranded cDNA was prepared from 300 ng RNA using the COVID Seq kit (Cat. No. 20051772). The RVOP library was prepared using Illumina DNA Prep (Illumina, Cat. No. 20044311). Agilent 2100 bioanalyzer was used to check the quality of both libraries. The RVOP library was denatured and diluted to optimal loading concentration for sequencing on the MiSeq platform using a v3 reagent kit at $2 \times 75$ bp read length. The sequencing data analysis was performed as previously described. The sequencing data analysis was performed as previously described[65–67].

### Minor variant analysis
The primer-free pair raw reads of SARSCoV2 were generated from the Illumina MiSeq. Raw reads of SARSCoV2 were pre-processed based on read quality and read length (phred quality ≥30 and minimum length ≥50 base pair) and merged by PEAR programme[68]. The merged reads were mapped to Wuhan's SARSCoV2 sequence (Genebank ID: NC_045512.2) to generate a consensus genome[69]. During the mapping of reads to reference genome sequence, a BAM file was generated by Samtools[70]. This BAM file was processed by diversiutils script in DiversiTools (http://josephhughes.github.io/btctools/) to find the frequency of all types of four bases for each position of a reference sequence. The only variants that have been covered by at least 15 times (read depth ≥15) by high-quality reads (average read's phred score ≥ 30) to find highly accurate single nucleotide variants (SNVs)[71].

### Statistical analysis
All the results were analysed and plotted using GraphPad Prism 8.0 software. Percentage change weight, relative gene expression, lung haemorrhagic scores, FACS, ELISA, and qPCR studies were compared and plotted as mean using graph pad. Dataset was analysed by using one-way ANOVA, two-way ANOVA, Wilcoxon test or Student's t-test. Differences were considered statistically significant with a p value of less than or equal to 0.05.

### Statistics and reproducibility
No statistical method was used to predetermine sample size. No data were excluded from the analyses. Mice of different genotypes were randomly assigned to treatment groups throughout the study. For experiments involving genetically modified animals, littermates were used for each experiment. In the cell and animal experiments, investigators were not blinded to group allocation because the investigators should give the drug to the mice and cell in different treatment conditions. Bio render software was used for pictorial representations.

### Reporting summary
Further information on research design is available in the Nature Portfolio Reporting Summary linked to this article.

## Data availability
The RNA sequence Data generated in this study has been deposited in the NCBI SRA database under the accession code no. PRJNA842504. Publicly available data with accession code, GSE209550. The authors declare that, the necessary data required to validate the findings of the paper can be found within the article itself or in the Supplementary Materials. Source data are provided with this paper.

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

## Acknowledgements

Funding support was provided to the AA laboratory from THSTI core grant, Translational Research Programme (TRP). We thank Dr. Bhabatosh Das, Dr. Pradipta, Dr. Shikha Saxena and Dr. Sweety Samal for helping with the sequencing and analysis of SARS-CoV-2 and B.1.1.529 in Next Generation Sequencing Lab at THSTI (New Delhi). We thank Dr. Sweety Samal, and Ritika Khatri for providing the virus. We thank FACS facility Incharge, Dr. Deepak Rathore for providing support. We acknowledge SAF and infectious disease research facility (IDRF) for its support. ILBS bio bank: for support in histological analysis and assessment. RCB microscopy facility: for microscopic examination of the histology slide. The following reagent was deposited by the Centers for Disease Control and Prevention and obtained through BEI Resources, NIAID, NIH: SARS Related Coronavirus 2, Isolate USA-WA1/2020, NR-52281, Isolate hCoV-19/USA/MD-HP20874/2021 (Lineage B.1.1.529, Omicron Variant), NR-56461, contributed by Andrew S. Pekosz, We acknowledge the BIRAC funding (BT/CS0054/21 and BT/CTH/0004/21) and the intramural funding from THSTI to support this study.

## Author contributions

Conceived, designed and supervised the study: A.A.; designed and performed the experiments: S.S.; ABSL3 experiment: S.S., R.D., J.D. and Z.A.R.; FACS: S.S., R.D. and V.D.; qPCR: S.S., R.D. and A.B.; ELISA: Z.A.R.; bright field microscopy imaging: S.S. and V.S.; genotyping: M.R.T. and S.G.; minor variant analysis: S.K.; analysed the Data: S.S.; contributed reagents/materials/analysis tools: A.A.; wrote the manuscript: S.S. and A.A.

## Competing interests

The authors declare no competing interests.
