## [Peer Review File · Nature Communications]

IL-9 aggravates SARS-CoV-2 infection and exacerbates associated airway inflammationREVIEWER COMMENTS

Reviewer #1 (Remarks to the Author):

Sadhu et al provide evidence that IL-9 exacerbates outcomes in a mouse model of SARS-CoV-2 infection. They show increased production/expression of IL-9 in PBMCs from infected patients (versus uninfected). Blocking IL-9 increases the anti-viral response and decreases viral pathologies. They show that blocking IL-9 can cooperate with remdesivir and that the Foxo1 transcription factor is required for IL-9 production in the T cell population.

Overall, this is an interesting paper that documents an as yet undescribed function of IL-9. The experiments are generally well done. There are a couple of experiments needed to substantiate some of the conclusions. Moreover, there is a scarcity of details in many of the studies that needs to be corrected.

1. One of the final experiments is to show that IL-9 administration can supplement the loss of Foxo1 in T cells. While important, this is a very crude experiment and it is doubtful that IL-9 injection mimics endogenous IL-9 production. I would suggest an additional experiment where the authors transfer WT SARS-CoV2-primed CD4 (or CD8) T cells to the Ace2 Tg Foxo1 fl/fl CD4-Cre mice. This would show that CD4 or CD8 T cells are sufficient as a source of IL-9 in the model. This would also support the underlying assumption that it is CD4 T cells supplying the IL-9. Have the authors examined CD8 T cell IL-9 production?

2. The authors need to be much clearer in the text and legends about the cells they are analyzing in each experiment and how they are treated. This was particularly a problem in describing data from patient samples. In the first sentence of the results the authors state "IL-9 levels were elevated in active COVID-19 patients infected...". It takes looking at the panel to realize IL-9 levels actually means IL9 mRNA and digging into the methods to see that this is mRNA from PBMCs. There is no description of the donors, no description of any stimulation of the cells, no description how long after infection the samples were taken. This needs to be described accurately and in more detail.

3. Some of the mast cell analysis is confusing. First, it seems like 3m and 3n should have some labels switched. The graph says mast cells but they seem to be gating on basophils, and vice versa. Second, the gating for CD117+FceRI+ cells in Fig. 3 seems very different than the gating in Fig. 2f and 5h.

4. The labels on Fig. 5d do not appear to be correct. Perhaps they should be the same as 5e?

5. The symbol legend shown in Fig. 2e belongs to Fig. 2d.

6. In Fig. 5f, the authors need to provide a clearer rationale for why the control mice and the cKO mice have histology examined on different days.

Reviewer #2 (Remarks to the Author):

Manuscript number: NCOMMS-22-28241-T

Title: "IL-9 exacerbates SARS-CoV2 infection and associated airway inflammation" by Amit Awasthi and colleagues.

In the present manuscript the authors provide sets of experimental approaches to demonstrate that T cell-derived IL-9 contributes to the pathophysiology evoked by SARS-CoV-2 infection and suggest IL-9 as a relevant therapeutic target to mitigate disease severity.

Employing different preclinical transgenic animal models, the authors dissect the role of IL-9 and Foxo1 in the regulation of SARS-CoV-2 infection and subsequent immune-mediated pathology. The present manuscript includes a great number of sophisticated experimental approaches to demonstrate a role of IL-9 in SARS-CoV-2 infection and immune-mediated disease severity.

However, some concerns rose during the review:

Major critique:

1. To understand the role of IL-9 in SARS-CoV-2 infection and immune-mediated pathology, the authors use transgenic mice expressing human ACE2 under control of the K18 promoter. In extended Data Fig 3i the authors demonstrate that IL-9 has a direct effect on hACE2 mRNA expression in this transgenic mouse model. Since hACE2 expression is under control of the human K18 promoter in this mouse model to direct its expression to epithelia and not physiologically regulated the authors need to rule out the possibility that some of the effects described in this manuscript are due to IL-9-dependent regulation of the transgenic K18 promoter activity.
2. It should be positively noted that the authors use many different preclinical model systems to analyze the role of IL-9 in SARS-CoV-2 infections. For this purpose, transcription factors, ISGs and the expression of antiviral genes are measured in addition to inflammatory cytokines. To maintain comparability of results, it would be useful to measure the expression of the same genes in all experiments. Is the expression of the genes analyzed directly regulated by IL-9/IL-9R signaling or indirectly?
3. Additionally, it is noticeable that in the introduction PU.1 is introduced as the master transcription factor for TH9 cells, but its expression was not analyzed at all.
4. Employing sophisticated experimental approaches, the authors identify mast cells, Eosinophils and basophils as important cell types responding to IL-9 upon SARS-CoV-2 infections. However, the contribution of these cell types to SARS-CoV-2 infection and immune-mediated pathology is not clear. Do these cells contribute to altered barrier function e.g. in the lungs of infected animals?
5. Immunity against SARS-CoV-2 seems to be mainly regulated by cytotoxic T cells and neutralizing antibodies. Is there an effect of IL-9 on cytotoxic T cells, B cells and/or antibody production?
6. Have the authors analyzed ILC-derived IL-9 as another important source of this cytokine?
7. The intracellular IL-9 staining in figure 2g looks quite strange. The authors should be encouraged to reanalyze the data.

Minor critique:

1. "SARS-CoV-2" is misspelled in the manuscript. A hyphen is missing between CoV and 2 in the pdf provided.
2. Writing and grammar should be checked throughout the manuscript.
3. In some passages, contradictory wording is chosen. E.g. in line 170 the authors write "mice were found to be resistant to SARS-CoV-2 infection" followed by "and survived infection better than...". If these mice are resistant to infection, I would not expect any disease symptoms or signs of infection.

Reviewer #3 (Remarks to the Author):

This manuscript by Sadhu and colleagues describes that the cytokine IL-9 exacerbates SARS-CoV-2 infection and airway inflammation. It is excellent work and the findings are very important to the field. I would like to add that the authors have done a tremendous amount of work on this manuscript, which it is extremely complete.

I have only a few small comments:

- For the introduction: Any published data from single cell studies on mast cells, eosinophils, and basophils that could be discussed to lines 51-54?
- Line 113. What are basal alveolar cells? A549 cells are human alveolar adenocarcinoma derived cells.
- Figure 2e. Add which is feces and which is lung.
- Discussion: Could you speculate on a mechanism by which IL-9 suppressed the IFN-I response?
- Most of the experimental data are from K18-ACE2 mice. Would other animal models lead to similar findings? A discussion section on this issue may be relevant.
- Were the SARS-CoV-2 stocks used for the experiments sequenced to confirm the absence of cell culture adaptations? If there are cell culture adaptations (generally in the multibasic cleavage site), please provide the percentages of each mutation.

REVIEWER COMMENTS

Reviewer #1 (Remarks to the Author):

Sadhu et al provide evidence that IL-9 exacerbates outcomes in a mouse model of SARS-CoV-2 infection. They show increased production/expression of IL-9 in PBMCs from infected patients (versus uninfected). Blocking IL-9 increases the anti-viral response and decreases viral pathologies. They show that blocking IL-9 can cooperate with remdesivir and that the Foxo1 transcription factor is required for IL-9 production in the T cell population.

Overall, this is an interesting paper that documents an as yet undescribed function of IL-9. The experiments are generally well done. There are a couple of experiments needed to substantiate some of the conclusions. Moreover, there is a scarcity of details in many of the studies that needs to be corrected.

Re: We thank the reviewer for the time he/she spent going through our manuscript and the comments which substantially improved the manuscript. We have provided the clarifications as and when needed and performed additional experiments as suggested.

1. One of the final experiments is to show that IL-9 administration can supplement the loss of Foxo1 in T cells. While important, this is a very crude experiment and it is doubtful that IL-9 injection mimics endogenous IL-9 production. I would suggest an additional experiment where the authors transfer WT SARS-CoV2-primed CD4 (or CD8) T cells to the Ace2 Tg Foxo1 fl/fl CD4-Cre mice. This would show that CD4 or CD8 T cells are sufficient as a source of IL-9 in the model. This would also support the underlying assumption that it is CD4 T cells supplying the IL-9. Have the authors examined CD8 T cell IL-9 production?

Re: We thank the reviewer for this critical point and suggestion, as the reviewer suggests to confirm that exogenous IL-9 mimics endogenous IL-9 production. More importantly reviewer wanted to confirm the source of IL-9. Since it was shown that IL-9 is primarily produced by CD4⁺ cells and innate lymphoid cells (ILCs) [1], we tested the cellular source of IL-9 in SARS-CoV2 infection in Ace2.Tg mice. First, we tested IL-9 cytokine staining in CD4⁺ T cell, NK cells and ILCs from uninfected and SARS-CoV-2 infected Ace2.Tg mice.

Our data suggest the IL-9 is primarily produced by CD4⁺ T cells; we did not find IL-9 production from NK cells and ILCs (**Extended Data Fig. 12 a-d**). We didn't see a significant difference in CD8⁺ IL-9⁺ T cells between uninfected and infected mice.

We further tested whether CD4⁺ T cell-driven IL-9 is sufficient in making Foxo1^{fl/fl}.CD4^{Cre+} mice susceptible to SARS-CoV-2 infection. To do this, we transferred the wt CD4⁺ T cells from ROSA mTmG mice into the Foxo1^{fl/fl}.CD4^{Cre+} mice; we referred these mice as wt-CD4T-Foxo1^{fl/fl}.CD4^{Cre+} mice. The advantage of using CD4⁺ T cells from ROSA mTmG mice, as these CD4⁺ T cells can be tracked *in vivo* based on their expression of RFP (Red fluorescence protein). Post transferring of CD4⁺ T cells from ROSA mTmG mice, we infected wt-CD4T-Foxo1^{fl/fl}.CD4^{Cre+} and Foxo1^{fl/fl}.CD4^{Cre+} mice with SARS-CoV-2 ancestral strain of SARS-CoV-2. We found that wt-CD4T-Foxo1^{fl/fl}.CD4^{Cre+} mice become as susceptible as wt Ace2.Tg mice while Foxo1^{fl/fl}.CD4^{Cre+} mice were found to remain less sensitive to SARS-CoV-2 infection. (**Extended data Fig. 12 e**). BALF analysis suggests that IL-9 is produced from wt mTmG, but not from Foxo1-deficient, CD4⁺ T cells, further confirming our findings that CD4⁺ T cell-derived IL-9 contributes to the susceptibility of SARS-CoV-2 infection (**Extended data Fig. 12 f-h**). We have incorporated these changes in the revised manuscript with highlighter (Line no. 261-281).

2. The authors need to be much clearer in the text and legends about the cells they are analyzing in each experiment and how they are treated. This was particularly a problem in describing data from patient samples. In the first sentence of the results the authors state "IL-9 levels were elevated in active COVID-19 patients infected...". It takes looking at the panel to realize IL-9 levels actually means IL9 mRNA and digging into the methods to see that this is mRNA from PBMCs. There is no description of the donors, no description of any stimulation of the cells, no description how long after infection the samples were taken. This needs to be described accurately and in more detail.

Re: We thank the reviewer for pointing out the missing information for more clarity for the readers. We incorporated all the information in the main text and methodology section of revised manuscript (**Supplementary table-01; line no.562-571**) and the detailed description of COVID-19 and Healthy individual details mentioned in **Supplementary table-01[2]**. The blood samples were collected as per the recommended guidelines of the Institutional Ethics Committee of THSTI (Human Research) and ESIC Hospital, Faridabad (Letter Ref No: THS 1.8.1/ (97) dated July 07, 2020). Blood samples were collected from symptomatic COVID-19 patients and healthy volunteers (n=9 each; **table S 1**) after receiving written informed consent (**methodology section line no.509-513**).

3. Some of the mast cell analysis is confusing. First, it seems like 3m and 3n should have some labels switched. The graph says mast cells but they seem to be gating on basophils, and vice versa. Second, the gating for CD117+FceRI⁺ cells in Fig. 3 seems very different than the gating in Fig. 2f and 5h.

Re: We thank the reviewer for spotting the error in labelling Fig.3m, 3n, Fig.2f, Fig.5h and basophil gating, which has been corrected in the revised manuscript.

4. The labels on Fig. 5d do not appear to be correct. Perhaps they should be the same as 5e?

Re: Thank you for spotting this error, we have corrected this in the revised manuscript.

5. The symbol legend shown in Fig. 2e belongs to Fig. 2d.

Re: We thank the reviewer for pointing out the mistake. We have removed the symbol legend. We have already shown the legend of the symbol in the x-axis itself.

6. In Fig. 5f, the authors need to provide a clearer rationale for why the control mice and the cKO mice have histology examined on different days.

Re: We thank the reviewer for this comment. Our data indicated that hACE2.Tg mice succumb to death within 7 to 8 days post-SARS-CoV-2 infection with severe lung pathologies (Fig. 2c,d) in the revised manuscript. Here the purpose of this experiment was different; as we wanted to test whether hAce2.Tg mice can transmit SARS-CoV-2 infection to healthy cohoused hAce2.Tg mice, compared to Foxofl/fl x CD4Cre+ transmit the infection to cohoused hAce2.Tg mice. Our previous data, indicate that Foxofl/fl x CD4Cre+ mice are less sensitive to SARS-CoV-2 infection compared to Foxofl/fl x CD4Cre- mice (Fig. 4a-b). This could be due to lesser viral replication in Foxofl/fl x CD4Cre+ mice. Keeping this in mind, we kept Foxofl/fl x CD4Cre+ mice cohoused with healthy hACE2.Tg mice for a longer period to see whether an extra given period allows an efficient SARS-CoV-2 transmission to healthy cohoused hACE2.Tg mice as it does in case of infected hACE2.Tg mice cohoused with healthy hACE2.Tg mice. Even though, providing this extra time to SARS-CoV-2 infected Foxofl/fl x CD4Cre+ mice were unable to transmit the infection, clearly indicating the inability Foxofl/fl x CD4Cre+ to transmit the infection. We incorporated the data in revised manuscript line no. 237-247.

Nonetheless, we agree with the reviewer that we should have compared between hACE2.Tg to hACE2.Tg and Foxofl/fl x CD4Cre+ to hACE2.Tg mice at the same time. Keeping this in mind, we have replaced the histology in **figure 5f** where we have the same time point comparison within the experimental group. We have kept the previous figure 5f as **Extended Data fig. 11** in the revised manuscript.

Reviewer #2 (Remarks to the Author):

Manuscript number: NCOMMS-22-28241-T

Title: “IL-9 exacerbates SARS-CoV2 infection and associated airway inflammation” by Amit Awasthi and colleagues.

In the present manuscript the authors provide sets of experimental approaches to demonstrate that T cell-derived IL-9 contributes to the pathophysiology evoked by SARS-CoV-2 infection and suggest IL-9 as a relevant therapeutic target to mitigate disease severity.

Employing different preclinical transgenic animal models, the authors dissect the role of IL-9 and Foxo1 in the regulation of SARS-CoV-2 infection and subsequent immune-mediated pathology.

The present manuscript includes a great number of sophisticated experimental approaches to demonstrate a role of IL-9 in SARS-CoV-2 infection and immune-mediated disease severity.

However, some concerns rose during the review:

Re: We thank the reviewer for the time he/she spent going through our manuscript and their comments which substantially improved the manuscript. We have provided the clarifications as and when needed and performed additional experiments as suggested.

Major critique:

1. To understand the role of IL-9 in SARS-CoV-2 infection and immune-mediated pathology, the authors use transgenic mice expressing human ACE2 under control of the K18 promoter. In extended Data Fig 3i the authors demonstrate that IL-9 has a direct effect on hACE2 mRNA expression in this transgenic mouse model. Since hACE2 expression is under control of the human K18 promoter in this mouse model to direct its expression to epithelia and not physiologically regulated the authors need to rule out the possibility that some of the effects

described in this manuscript are due to IL-9-dependent regulation of the transgenic K18 promoter activity.

Re: We thank the reviewer for raising a valid point. However, it was shown that Th2 cytokines, particularly IL-13, was shown to enhance ACE2 expression, whereas IL-17 also enhanced ACE2 expression in the case of human [3, 4]. Expression of hACE2 is directly related to the degree of SARS-CoV-2 infection in different tissues like the Lungs, Intestine, and heart [5]. The protein expression levels of ACE2 was relatively higher in the small intestine than in other tissues of human [6]. However, it is not known whether IL-9 enhances ACE2 expression. It is very important to find out whether IL-9 regulates the expression of ACE2 under K18 promoter. In our opinion, this point is beyond the scope of this manuscript; therefore, we would like to remove ACE2 expression data in the revised manuscript and might use it in subsequent manuscript to identify whether ACE2 expression is modulated by Th2 and other cytokines.

It is important to note that expression of IL-9 increases between 2-4 day post SARS-CoV-2 infection in both ACE2.Tg mice and hamster. By this time the SARS-CoV-2 infection found to be established in these two models, indicating that the IL-9-mediated regulation of ACE2 may not be important to establishing the infection and associated effects in ACE2.Tg mice.

2. It should be positively noted that the authors use many different preclinical model systems to analyze the role of IL-9 in SARS-CoV-2 infections. For this purpose, transcription factors, ISGs and the expression of antiviral genes are measured in addition to inflammatory cytokines. To maintain comparability of results, it would be useful to measure the expression of the same genes in all experiments. Is the expression of the genes analyzed directly regulated by IL-9/IL-9R signalling or indirectly?

Re: We thank the reviewer for the suggestion. We have measured the same set of genes, like transcription factors, ISGs, and the expression of anti-viral genes in addition to inflammatory cytokines in all the settings. By keeping the reviewer's suggestion in mind, In fig.3, we performed qRT-PCR for ADAR; we found significant upregulation of ADAR compared to Infection (data added in **fig.3 r**; and incorporated the cGAS and STING genes data in **extended data fig.6 i-j**). We found that RNase L and ADAR gene expression are missing in the presence of rIL-9 treatment. Therefore, we also performed qRT-PCR for these genes and found that relative ADAR expression was significantly downregulated in the presence of IL9 (**Extended Data Fig. 3f**). Still, we haven't found a significant difference in RNase L. Therefore, we have incorporated the ADAR data in revised manuscript (**extended data fig.3f**) but not RNase L.

Although there is no clear mechanism is identified as to how IL-9-meidates suppression of type 1 IFN response. Our data indicated that while anti-IL-9 antibody neutralization suppressed SARS-CoV-2 infection and increased anti-viral type 1 IFN response, exogenous recombinant IL-9 treatment enhanced SARS-CoV-2 infection and suppressed anti-viral type 1 IFN response. Upon binding to its receptor, IL-9 activates proximal signalling events primarily driven by JAK-STAT pathways, which includes JAK1, JAK3 and STAT1, STAT3 and STAT5[7]. Antiviral response of type 1 IFNs also requires activation of STAT proteins. It was shown that STAT3 negatively regulates type 1 IFNs response, as STAT3 deficient cells produced enhanced type 1 IFNs, which in turn increased ISGs expression[8]. Since IL-9 activates STAT3, it might suppress type 1 IFNs production its subsequent function in ISGs expression. This is could be one of the mechanisms by which IL-9 may suppress type 1 IFNs response. However, this needs to be experimentally evaluated (Incorporated the text in discussion section of revised manuscript **line no.355-366**).

3. Additionally, it is noticeable that in the introduction PU.1 is introduced as the master transcription factor for TH9 cells, but its expression was not analyzed at all.

Re: We thank the reviewer for noticing the lack of PU.1 expression data in all these experiments. Therefore we performed the qRT-PCR for relative PU.1 (SPI1) expression. We found that, the expression of SPI1 significantly decreased in the presence of Foxo1 (Extended data figure.6b right panel), and Foxo1^{fl/fl}xCD4^{Cre+} (Extended data figure.8b) mice compared to infection. We have incorporated these data in revised manuscript.

4. Employing sophisticated experimental approaches, the authors identify mast cells, Eosinophils and basophils as important cell types responding to IL-9 upon SARS-CoV-2 infections. However, the contribution of these cell types to SARS-CoV-2 infection and immune-mediated pathology is not clear. Do these cells contribute to altered barrier function e.g. in the lungs of infected animals?

Re: Here we described that Eosinophils, Mast cells, and Basophils are allergy-inducing cells essential in causing lung inflammation in asthma. In COVID-19 settings, SARS-CoV-2 causes acute lung injury and inflammation with some degree of similarity with allergic inflammation. IL-9 produced in SARS-CoV-2 infection recruit and promotes the growth of mast cells, which in turn enhances IL-9-mediated lung inflammation. We did not perform any lung epithelial barrier function experiments in the presence of IL-9 due to added complexity of handling infected tissue in Bsl3 facility. Nonetheless it is well established that SARS-CoV-2 causes lung injury, it damages the epithelial barrier and alters its function [9, 10]. In asthma, bronchitis, and Allergic lung diseases, eosinophils, mast cells, and basophils play an essential role in lung inflammation by enhancing mucus production through increased expression of IL-13, and IL-5 [11-13]. In line with this, we also see accumulation of eosinophils, mast cells, and basophils, which may contribute lung inflammation.

5. Immunity against SARS-CoV-2 seems to be mainly regulated by cytotoxic T cells and neutralizing antibodies. Is there an effect of IL-9 on cytotoxic T cells, B cells and/or antibody production?

Re: This is very important point reviewer has raised. We did not test SARS-CoV-2 specific Ab in this setting as the hACE2.Tg mice model is acute model in which mice succumb to death between 6-8 days. This time is not sufficient for the generation of high affinity antibodies, which takes more than 14 or so days. Moreover, here we wanted to understand the mechanism of IL-9 mediated pathologies and inflammation, and as far as we know the role of B cell and antibodies are not been associated with inflammation and lung pathologies. Similarly, in this model we did not see any changes in CD8 frequency in this setting.

6. Have the authors analyzed ILC-derived IL-9 as another important source of this cytokine?

Re: We thank the reviewer for the critical suggestion. Other than CD4, NK and ILCs are the vital sources of IL-9; therefore, we performed the in-vivo experiment to determine the role of these cells. We didn't find any IL-9 production from in NK cells and ILC cells post SARS-CoV-2 infection (Extended Data Fig. 12 a-d).

We further tested whether CD4+ T cell-driven IL-9 is sufficient in making Foxo1^{fl/fl}.CD4^{Cre+} mice susceptible to SARS-CoV-2 infection. To do this, we transferred the wt CD4+ T cells from ROSA mTmG mice into the Foxo1^{fl/fl}.CD4^{Cre+} mice; we referred these mice as wt-CD4T-Foxo1^{fl/fl}.CD4^{Cre+} mice. The advantage of using CD4+ T cells from ROSA mTmG mice, as these CD4+ T cells can be tracked *in vivo* based on their expression of RFP (Red fluorescence protein). Post transferring of CD4+ T cells from ROSA mTmG mice, we infected wt-CD4T-Foxo1^{fl/fl}.CD4^{Cre+} and Foxo1^{fl/fl}.CD4^{Cre+} mice with SARS-CoV-2 ancestral strain of SARS-CoV-2. We found that wt-CD4T-Foxo1^{fl/fl}.CD4^{Cre+} mice become as susceptible as wt Ace2.Tg mice while Foxo1^{fl/fl}.CD4^{Cre+} mice were found to be remain resistant to SARS-CoV-2 infection. (Extended data Fig. 12 e). BALF analysis suggest that

IL-9 is produced from wt mTmG, but not from Foxo1-deficient, CD4⁺ T cells, further confirming our findings that CD4⁺ T cell-derived IL-9 contributes to the susceptibility of SARS-CoV-2 infection (**Extended data Fig. 12 f-h**). We have incorporated these changes in the revised manuscript with highlighter (**Line no.261-281; Discussion section line no. 337-349**).

7. The intracellular IL-9 staining in figure 2g looks quite strange. The authors should be encouraged to reanalyze the data.

Re: We thank the reviewer for the suggestion to reanalyse the data. The FACS plot might look strange due to the cell number of CD4 T cells is less in number. Therefore we have removed the BALF data and incorporated the data of draining lymph nodes of the Lungs (mediastinal and brachial lymph nodes), which resembles the immune response in BALF.

Minor critique:

1. "SARS-CoV-2" is misspelled in the manuscript. A hyphen is missing between CoV and 2 in the pdf provided.

Re: We rectified in the revised manuscript and highlighted.

2. Writing and grammar should be checked throughout the manuscript.

Re: We have reviewed the whole manuscript and rectified the grammar and spelling wherever necessary.

3. In some passages, contradictory wording is chosen. E.g. in line 170 the authors write "mice were found to be resistant to SARS-CoV-2 infection" followed by "and survived infection better than...". If these mice are resistant to infection, I would not expect any disease symptoms or signs of infection.

Re: We thank reviewer for correcting the word use. We changed the word resistant to less sensitive, as we can see the mice remained healthy with no sign of weight loss, and survived infection better than RDV treatment without substantial lung lesions and lung viral load.

Reviewer #3 (Remarks to the Author):

This manuscript by Sadhu and colleagues describes that the cytokine IL-9 exacerbates SARS-CoV-2 infection and airway inflammation. It is excellent work and the findings are very important to the field. I would like to add that the authors have done a tremendous amount of work on this manuscript, which it is extremely complete.

I have only a few small comments:

- For the introduction: Any published data from single cell studies on mast cells, eosinophils, and basophils that could be discussed to lines 51-54?

Re: Earlier literature suggests that Mast cells and Eosinophils are innate immune cells that play pathogenic roles in many inflammatory responses [14]. Mast cell-derived proteases and eosinophil-associated mediators are elevated in COVID-19 patient sera and lung tissues[14]. Mast cell activation in humans was confirmed by detecting the Mast cell-specific protease and chymase, which significantly correlated with disease severity. These results support the association of Mast cell activation with severe COVID-19[15]. We have incorporated these references in the revised manuscript (line no.55-59).

- Line 113. What are basal alveolar cells? A549 cells are human alveolar adenocarcinoma derived cells.

Re: We have rectified the mistake and highlighted in the manuscript as “human alveolar adenocarcinoma derived epithelial cells”.

- Figure 2e. Add which is feces and which is lung.

Re: We have added the sample names in the revised manuscript (Fig.2e)

-Discussion: Could you speculate on a mechanism by which IL-9 suppressed the IFN-I response?

Re: Although the mechanism of IL-9-mediated suppression of type 1 IFN response is not established yet. Our data indicated that while anti-IL-9 antibody neutralization suppressed SARS-CoV-2 infection and increased anti-viral type 1 IFN response, exogenous recombinant IL-9 treatment enhanced SARS-CoV-2 infection and suppressed anti-viral type 1 IFN response. Upon binding to its receptor, IL-9 activates proximal signalling events primarily driven by JAK-STAT pathways, which includes JAK1, JAK3 and STAT1, STAT3 and STAT5 [7]. Antiviral response of type 1 IFNs also requires activation of STAT proteins. It was shown that STAT3 negatively regulates type 1 IFNs response, as STAT3 deficient cells produced enhanced type 1 IFNs, which in turn increased ISGs expression [8]. Since IL-9 activates STAT3, it might suppress type 1 IFNs production its subsequent function in ISGs expression. This could be one of the mechanisms by which IL-9 may suppress type 1 IFNs response. However, this needs to be experimentally evaluated. We have included this in the discussion part in revised manuscript (**line 355-366**).

- Most of the experimental data are from K18-ACE2 mice. Would other animal models lead to similar findings? A discussion section on this issue may be relevant.

Re: We thank the reviewer for the suggestion. Due to availability of mouse reagents, we mechanistically studied the role of IL-9 in this hAce2.Tg model. Nonetheless, we do see an increase in mRNA expression in PBMC from active COVID19 patients and SARS-CoV-2 infected hamster. However, we found that IL-9 levels were upregulated in Human, and Hamster model. In line with this, we speculate that IL-9 might be important in causing lung inflammation and pathologies in other models. We have included this in the discussion in the revised version of manuscript (**line 318-324**).

- Were the SARS-CoV-2 stocks used for the experiments sequenced to confirm the absence of cell culture adaptations? If there are cell culture adaptations (generally in the multibasic cleavage site), please provide the percentages of each mutation.

Re: We thank the reviewer for the valuable suggestion to check the any cell culture adapted mutations. We have performed the sequencing of the ancestral SARS-CoV-2 and B.1.1.529 stocks we have used and the sequencing results showed no mutation. The sequencing data included in the revised manuscript as **Supplementary table. 2-3** and in the methodology section **line no.743-756**.

1. Wilhelm, C., et al., *An IL-9 fate reporter demonstrates the induction of an innate IL-9 response in lung inflammation*. Nat Immunol, 2011. **12**(11): p. 1071-7.
2. Thiruvengadam, R., et al., *Effectiveness of ChAdOx1 nCoV-19 vaccine against SARS-CoV-2 infection during the delta (B.1.617.2) variant surge in India: a test-negative, case-control study and a mechanistic study of post-vaccination immune responses*. Lancet Infect Dis, 2022. **22**(4): p. 473-482.
3. Kimura, H., et al., *Type 2 inflammation modulates ACE2 and TMPRSS2 in airway epithelial cells*. J Allergy Clin Immunol, 2020. **146**(1): p. 80-88.e8.

4. Branco, A., M.N. Sato, and R.W. Alberca, *The Possible Dual Role of the ACE2 Receptor in Asthma and Coronavirus (SARS-CoV2) Infection*. Front Cell Infect Microbiol, 2020. **10**: p. 550571.
5. Dong, W., et al., *The K18-Human ACE2 Transgenic Mouse Model Recapitulates Non-severe and Severe COVID-19 in Response to an Infectious Dose of the SARS-CoV-2 Virus*. Journal of Virology, 2022. **96**(1): p. e00964-21.
6. Zhang, H., et al., *Specific ACE2 expression in small intestinal enterocytes may cause gastrointestinal symptoms and injury after 2019-nCoV infection*. Int J Infect Dis, 2020. **96**: p. 19-24.
7. Goswami, R. and M.H. Kaplan, *A brief history of IL-9*. Journal of immunology (Baltimore, Md. : 1950), 2011. **186**(6): p. 3283-3288.
8. Wang, W.B., D.E. Levy, and C.K. Lee, *STAT3 negatively regulates type I IFN-mediated antiviral response*. J Immunol, 2011. **187**(5): p. 2578-85.
9. D'Agnillo, F., et al., *Lung epithelial and endothelial damage, loss of tissue repair, inhibition of fibrinolysis, and cellular senescence in fatal COVID-19*. Science Translational Medicine, 2021. **13**(620): p. eabj7790.
10. Wu, M.-L., et al., *SARS-CoV-2-triggered mast cell rapid degranulation induces alveolar epithelial inflammation and lung injury*. Signal Transduction and Targeted Therapy, 2021. **6**(1): p. 428.
11. Lacoste, J.-Y., et al., *Eosinophilic and neutrophilic inflammation in asthma, chronic bronchitis, and chronic obstructive pulmonary disease*. Journal of allergy and clinical immunology, 1993. **92**(4): p. 537-548.
12. Carlier, F.M., C. de Fays, and C. Pilette, *Epithelial Barrier Dysfunction in Chronic Respiratory Diseases*. Front Physiol, 2021. **12**: p. 691227.
13. Nguyen, N., et al., *TGF- β 1 alters esophageal epithelial barrier function by attenuation of claudin-7 in eosinophilic esophagitis*. Mucosal Immunology, 2018. **11**(2): p. 415-426.
14. Gebremeskel, S., et al., *Mast Cell and Eosinophil Activation Are Associated With COVID-19 and TLR-Mediated Viral Inflammation: Implications for an Anti-Siglec-8 Antibody*. Frontiers in Immunology, 2021. **12**.
15. Tan, J., et al., *Signatures of mast cell activation are associated with severe COVID-19*. medRxiv, 2021.

REVIEWER COMMENTS

Reviewer #1 (Remarks to the Author):

The authors have addressed my previous concerns and added additional data to manuscript.

Two minor points that were not addressed.

1. As noted previously, the population assessed as mast cells in 3m looks very different than in 5h or 2f. Is there a reason for this? Different gating?
2. The authors provided more information on the patient samples in Fig. 1a as a supplementary table. However, they did not address the previous criticism that it was not stated that mRNA was from PBMCs in the figure legend. The source of mRNA is an important point.

Reviewer #2 (Remarks to the Author):

In the revised version of their manuscript, the authors have satisfactorily addressed most of my concerns.

Unfortunately, however, an important and central question remains unanswered, namely whether IL-9 regulates the activity of the transgenic K18 promoter and in turn regulates the expression of the transgenic hACE2 in a non-physiological manner in the model used.

In my opinion, the authors should be encouraged to exclude that many of the findings shown in the manuscript are due to non-physiological/K18 promoter-dependent regulation of hACE2 expression by IL-9.

Reviewer #3 (Remarks to the Author):

I am satisfied with all revisions and changes except one.

I previously asked for the authors to report the sequence data of the isolates they used, because the virus culture method that was used by the authors can lead to adaptations in the multi basic cleavage site, or S1/S2 site. See my question and the response by the authors below:

"- Were the SARS-CoV-2 stocks used for the experiments sequenced to confirm the absence of cell culture adaptations? If there are cell culture adaptations (generally in the multibasic cleavage site), please provide the percentages of each mutation.

Re: We thank the reviewer for the valuable suggestion to check the any cell culture adapted mutations. We have performed the sequencing of the ancestral SARS-CoV-2 and B.1.1.529 stocks we have used and the sequencing results showed no mutation. The sequencing data included in the revised manuscript as Supplementary table. 2-3 and in the methodology section line no.743-756."

I checked the sequence data provided by the authors. The virus isolate from supp. table 2 seems to have mutations in the S1/S2 cleavage site, specifically at amino acid 680. This codon is supposed to code for an arginine that is critical for efficient cleavage. Isolates with a mutated arginine at this site are attenuated in vivo (PMID 32821033, 33835028, 33494095). In the sequence provided in supp. table 2, the sequence for this codon is YKG. The ambiguous nucleotides Y and K seem to indicate a mix of nucleotides at position 1 and 2 of this codon. The isolate in supp. table 3 has the same issue but with one mutation instead of two in the same codon, leading to a YGG codon.

The authors should correct their submitted sequences to actually consensus sequences, showing the dominant nucleotide at every position. It is misleading to use ambiguous nucleotide like Y and K because these do not show up as mismatched/mutations when performing in sequence comparison algorithms.

The authors should mention in the manuscript whether or not they worked with a virus stock that contained S1/S2 cleavage site mutations, and specify which % of their virus stock is mutated and which % is wild type virus. They should discuss briefly how this may have affected the results of the paper, based on the percentages of wild type and mutant viruses. This can be done briefly in the methods section. Other mutations that are now hidden because of the use of ambiguous nucleotides should also be reported.

Lastly, there is an issue with the isolate from supp. table 3. The first ~2000 bp are missing from the genome.

Reviewer #1 (Remarks to the Author):

The authors have addressed my previous concerns and added additional data to manuscript.

Two minor points that were not addressed.

1. As noted previously, the population assessed as mast cells in 3m looks very different than in 5h or 2f. Is there a reason for this? Different gating?

Re: We thank the reviewer for raising this point. We would like to clarify that we used similar gating strategy for the mast cells throughout the manuscript, and by keeping the reviewer comment in mind, we reanalysed the file and rectified the gate of the mast cells in Fig.3M. We incorporated these changes in revised manuscript.

2. The authors provided more information on the patient samples in Fig. 1a as a supplementary table. However, they did not address the previous criticism that it was not stated that mRNA was from PBMCs in the figure legend. The source of mRNA is an important point.

Re: We included the source of mRNA in the methodology section stating as “isolated the total RNA from PBMC using the Trizol reagent from active COVID19 patients’ blood sample, and isolated RNA was used to test the expression of respective genes. By keeping the reviewer point in mind, we incorporated the mRNA source in the figure legend as well (Line no. 435-436)

Reviewer #2 (Remarks to the Author):

In the revised version of their manuscript, the authors have satisfactorily addressed most of my concerns. Unfortunately, however, an important and central question remains unanswered, namely whether IL-9 regulates the activity of the transgenic K18 promoter and in turn regulates the expression of the transgenic hACE2 in a non-physiological manner in the model used.

In my opinion, the authors should be encouraged to exclude that many of the findings shown in the manuscript are due to non-physiological/K18 promoter-dependent regulation of hACE2 expression by IL-9.

Re: We thank the reviewer for raising this critical point. To address the role of IL-9 in regulating the activity of the transgenic K18 promoter, this might regulate the expression of the transgenic hACE2. To address this point, we performed Western Blot experiment to test the expression of hACE2 at protein level in the presence or absence of IL-9 treatment *in vitro* and *in vivo*. Briefly, K18hAce2 mice were administered rIL-9 (500ng/mice, the similar dose were used in the manuscript Fig 1) or vehicle intranasally for 24 hrs before euthanising the mice. Lungs from these were homogenised in protein lysis buffer for Western Blot analysis using anti-human Ace2 antibody. Our data indicated that IL-9 treatment did not increase, as compared to control treatment, hAce2 expression in K18hAce2.tg mice (Extended Data Fig. 5g in revised manuscript).

To further confirm this data in human settings, we used the epithelial cell line, Caco2, which is known to express Ace2 [1, 2]. Briefly, Caco2 cell line was treated with recombinant human IL-9 (10ng/ml) for 24 hrs. Post IL-9 treatment, these treated cells were lysed with protein lysis buffer for further Western Blot analysis. Our result indicate that IL-9 treatment did not enhance Ace2 expression in Caco2 cells (Extended Data Fig. 5g in revised manuscript) further substantiating our findings that IL-9 does not increase expression of hAce2 in K18Ace2.tg mice, and thus does not regulate the expression of the transgenic hACE2 in a non-physiological manner IL-9.

Reviewer #3 (Remarks to the Author):

I am satisfied with all revisions and changes except one.

I previously asked for the authors to report the sequence data of the isolates they used, because the virus culture method that was used by the authors can lead to adaptations in the multi basic cleavage site, or S1/S2 site. See my question and the response by the authors below:

"- Were the SARS-CoV-2 stocks used for the experiments sequenced to confirm the absence of cell culture adaptations? If there are cell culture adaptations (generally in the **multibasic cleavage site**), please provide the percentages of each mutation.

Re: We thank the reviewer for the valuable suggestion to check the any cell culture adapted mutations. We have performed the sequencing of the ancestral SARS-CoV-2 and B.1.1.529 stocks we have used and the sequencing results showed no mutation. The sequencing data included in the revised manuscript as Supplementary table. 2-3 and in the methodology section line no.743-756."

I checked the sequence data provided by the authors. The virus isolate from supp. table 2 seems to have mutations in the S1/S2 cleavage site, specifically at amino acid 680. This codon is supposed to code for an arginine that is critical for efficient cleavage. Isolates with a mutated arginine at this site are attenuated in vivo (PMID 32821033, 33835028, 33494095). In the sequence provided in supp. table 2, the sequence for this codon is YKG. The ambiguous nucleotides Y and K seem to indicate a mix of nucleotides at position 1 and 2 of this codon. The isolate in supp. table 3 has the same issue but with one mutation instead of two in the same codon, leading to a YGG codon.

The authors should correct their submitted sequences to actually consensus sequences, showing the dominant nucleotide at every position. It is misleading to use ambiguous nucleotide like Y and K because these do not show up as mismatched/mutations when performing in sequence comparison algorithms.

The authors should mention in the manuscript whether or not they worked with a virus stock that contained S1/S2 cleavage site mutations, and specify which % of their virus stock is mutated and which % is wild type virus. They should discuss briefly how this may have affected the results of the paper, based on the percentages of wild type and mutant viruses. This can be done briefly in the methods section. Other mutations that are now hidden because of the use of ambiguous nucleotides should also be reported.

Lastly, there is an issue with the isolate from supp. table 3. The first ~2000 bp are missing from the genome.

Re: We thank the reviewer for the valuable time to spent analyzing the sequence. To avoid ambiguity in the sequence, as many nucleotides were undetected and the data quality was not as clean. Therefore, we performed the sequencing again for both ancestral and Omicron stains of SARS-CoV2. We reanalysed the sequencing results and aligned consensus sequence of ancestral and Omicron strains. Aligned spike nucleotide of both ancestral and Omicron with consensus, we found that there is no mutations at amino acid 683 in the spike region of both the Wuhan and Omicron (S1/S2 cleavage site) viruses compared to the original consensus sequence. Moreover, we found Serine at 680 instead of Arginine in our sequence analysis, as this the case in original consensus sequence. We incorporated these changes (bioinformatics analysis results) in the revised manuscript (Extended Data Fig.1a-b; Supplementary Table. 2 and 3).

In the earlier submitted sequence, first ~2000 bp were missing from the Omicron genome, however, in the resequencing data, we could able to cover these missing nucleotides in Omicron.

1. Mossel, E.C., et al., *Exogenous ACE2 expression allows refractory cell lines to support severe acute respiratory syndrome coronavirus replication*. J Virol, 2005. **79**(6): p. 3846-50.
2. Sherman, E.J. and B.T. Emmer, *ACE2 protein expression within isogenic cell lines is heterogeneous and associated with distinct transcriptomes*. Scientific Reports, 2021. **11**(1): p. 15900.

REVIEWER COMMENTS

Reviewer #1 (Remarks to the Author):

The authors have addressed the previous concerns.

Reviewer #2 (Remarks to the Author):

In the revised version of their manuscript, the authors have satisfactorily addressed my final concerns. I would therefore like to recommend the manuscript for publication.

Reviewer #3 (Remarks to the Author):

I thank the authors for their response and for noticing my typo. 680 should have indeed been 683. I notified the editor of this typo, which I hope came through.

The authors mention that they aligned the consensus sequences to the respective references. This is not what I asked. With the analysis that is done now, it could be for example that 40% of the virus stock is mutated (this would not show up in the consensus). The ambiguous nucleotides at codon 683 suggest a mixture of different nucleotides. Therefore, I asked the authors to provide "which % of their virus stock is mutated and which % is wild type virus". To be clear, I am asking for a minor variant analysis of their stocks, for example in a table format, showing the minor variants (for example down to 10%) detected at each site of the genome. The question is then whether there are minor variant that cripple the virus, like mutations at site 683, and these need to be mentioned.

The authors have not provided this data and therefore I am not satisfied with this response, unfortunately, because this manuscript, I repeat, is elegant work and important to the field.

REVIEWER COMMENTS

Reviewer #1 (Remarks to the Author):

The authors have addressed the previous concerns.

Re: We thank the reviewer for the valuable time to review our manuscript and accepting for the publication.

Reviewer #2 (Remarks to the Author):

In the revised version of their manuscript, the authors have satisfactorily addressed my final concerns. I would therefore like to recommend the manuscript for publication.

Re: We thank the reviewer for the valuable time to review our manuscript and accepting for the publication.

Reviewer #3 (Remarks to the Author):

I thank the authors for their response and for noticing my typo. 680 should have indeed been 683. I notified the editor of this typo, which I hope came through.

The authors mention that they aligned the consensus sequences to the respective references. This is not what I asked. With the analysis that is done now, it could be for example that 40% of the virus stock is mutated (this would not show up in the consensus). The ambiguous nucleotides at codon 683 suggest a mixture of different nucleotides. Therefore, I asked the authors to provide "which % of their virus stock is mutated and which % is wild type virus". To be clear, I am asking for a minor variant analysis of their stocks, for example in a table format, showing the minor variants (for example down to 10%) detected at each site of the genome. The question is then whether there are minor variant that cripple the virus, like mutations at site 683, and these need to be mentioned.

The authors have not provided this data and therefore I am not satisfied with this response, unfortunately, because this manuscript, I repeat, is elegant work and important to the field.

Re: We thank the reviewer for his/her valuable time to review the manuscript. We apologise for misunderstanding of the comment raised by the reviewer. Therefore, we have removed the alignment data instead incorporating the minor variant analysis (Supplementary Table 2) in the revised manuscript as suggested.

Minor variant analysis indicated the presence of five SNVs in ancestral and twenty four in Omicron strains of SARS-CoV-2. None of these SNVs are found to be in the cleavage site of virus, therefore it is highly unlikely to affect infectivity, lethality of the virus or weaken the virus's pathogenicity. This is supported by the experimental evidences we provided in the manuscript in which we found that infection of hACE2.Tg mice with ancestral strain of SARS-CoV2 causes lethality and mice become moribund post infection as reported earlier [1, 2]. As

reported earlier, Omicron strain causes attenuated infection in hAce2.Tg mice [3]. The details of each SNVs were mentioned in Supplementary table 2. We incorporated the changes in revised manuscript (line no. 85-94 &126-128).

Here is methodology we used for Minor variant analysis. Briefly, the primer-free pair raw reads of SARSCoV2 were generated from the Nano pore Seq. Raw reads of SARSCoV2 were pre-processed based on read quality and read length (phred quality ≥ 30 and minimum length ≥ 50 base pair) and merged by PEAR program [4]. The merged reads were mapped to Wuhan's SARSCoV2 sequence (Genebank ID: NC_045512.2) to generate a consensus genome [5]. During the mapping of reads to reference genome sequence, a BAM file was generated by Same tools [6]. This BAM file was processed by diversi script in Diversi Tools (<http://josephhughes.github.io/btctools/>) to find the frequency of all types of four bases for each position of a reference sequence. The only variants that have been covered by at least 15 times (read depth ≥ 15) by high-quality reads (average read's phred score ≥ 30) to find highly accurate single nucleotide variants (SNVs) [7].

1. Bao, L., et al., *The pathogenicity of SARS-CoV-2 in hACE2 transgenic mice*. Nature, 2020. **583**(7818): p. 830-833.
2. Winkler, E.S., et al., *SARS-CoV-2 infection of human ACE2-transgenic mice causes severe lung inflammation and impaired function*. Nat Immunol, 2020. **21**(11): p. 1327-1335.
3. Halfmann, P.J., et al., *SARS-CoV-2 Omicron virus causes attenuated disease in mice and hamsters*. Nature, 2022. **603**(7902): p. 687-692.
4. Zhang, J., et al., *PEAR: a fast and accurate Illumina Paired-End reAd merger*. Bioinformatics, 2014. **30**(5): p. 614-20.
5. Poojary, M., et al., *Computational Protocol for Assembly and Analysis of SARS-nCoV-2 Genomes*. 2020: p. 1-14.
6. Li, H., et al., *The Sequence Alignment/Map format and SAMtools*. Bioinformatics, 2009. **25**(16): p. 2078-2079.
7. Song, K., L. Li, and G. Zhang, *Coverage recommendation for genotyping analysis of highly heterologous species using next-generation sequencing technology*. Scientific Reports, 2016. **6**(1): p. 35736.

REVIEWERS' COMMENTS

Reviewer #3 (Remarks to the Author):

I would like to thank the authors for their response. My concerns have been addressed, but I do not understand the column labels in the table and why they are different for SARS-CoV-2 WT and Omicron.

I would suggest to amend the SARS-CoV-2 table to follow the same format as Omicron with the indication for the amounts of reads containing and A, C T or G.

Adding a legend would help as well.

When this amendment is made I do not need to see the manuscript again.

REVIEWERS' COMMENTS

Reviewer #3 (Remarks to the Author):

I would like to thank the authors for their response. My concerns have been addressed, but I do not understand the column labels in the table and why they are different for SARS-CoV-2 WT and Omicron.

I would suggest to amend the SARS-CoV-2 table to follow the same format as Omicron with the indication for the amounts of reads containing and A, C T or G.

Adding a legend would help as well.

When this amendment is made I do not need to see the manuscript again.

Re: We express our gratitude to the reviewer for dedicating his/her valuable time to review our manuscript and for providing insightful suggestions. In line with the reviewer's feedback, we have ensured that the column labels in both the SARS-CoV-2 and Omicron datasets are maintained in the same format and clearly indicate the amount of reads containing A, T, G, and C with proper labels.